# Unveiling m-Sharpness Through the Structure of Stochastic Gradient Noise

**Haocheng Luo**     **Mehrtash Harandi**     **Dinh Phung**     **Trung Le**
Monash University, Australia
{haocheng.luo, mehrtash.harandi, dinh.phung, trunglm}@monash.edu

## Abstract

Sharpness-aware minimization (SAM) has emerged as a highly effective technique to improve model generalization, but its underlying principles are not fully understood. We investigate m-sharpness, where SAM performance improves monotonically as the micro-batch size for computing perturbations decreases, a phenomenon critical for distributed training yet lacking rigorous explanation. We leverage an extended Stochastic Differential Equation (SDE) framework and analyze stochastic gradient noise (SGN) to characterize the dynamics of SAM variants, including n-SAM and m-SAM. Our analysis reveals that stochastic perturbations induce an implicit variance-based sharpness regularization whose strength increases as m decreases. Motivated by this insight, we propose Reweighted SAM (RW-SAM), which employs sharpness-weighted sampling to mimic the generalization benefits of m-SAM while remaining parallelizable. Comprehensive experiments validate our theory and method.

## 1 Introduction

In machine learning, gradient-based optimization algorithms aim to minimize the following loss function:

$$\min_{x \in \mathbb{R}^d} f(x) := \frac{1}{n} \sum_{i=1}^{n} f_i(x), \tag{1}$$

where $x \in \mathbb{R}^d$ denotes the parameter, $f_i(x)$ represents the loss on the $i$-th sample, $i$ ranges from 1 to $n$, and $n$ is the size of the training set. We primarily focus on stochastic algorithms, where at each step $k$, an index set $\gamma_k$ of fixed cardinality $|\gamma|$ is sampled uniformly at random. We denote the mini-batch loss by $f_{\gamma_k}(x) = \frac{1}{|\gamma|} \sum_{i \in \gamma_k} f_i(x)$.

We investigate the recently proposed Sharpness-Aware Minimization (SAM) (Foret et al., 2021), which has achieved remarkable success in various application domains (Foret et al., 2021; Kwon et al., 2021; Kaddour et al., 2022; Liu et al., 2022a; Qu et al., 2022; Wang et al., 2024; Nguyen et al., 2024; Hoang-Anh et al., 2025; Singh et al., 2025). It seeks flat minima by minimizing the perturbed loss $\min_{x \in \mathbb{R}^d} f(x + \rho \, \epsilon^*(x))$, where the perturbation $\epsilon^*(x)$ is defined as the solution to

$$\max_{\|\epsilon\| \leq 1} \langle \nabla f(x), \epsilon \rangle, \tag{2}$$

which admits the closed-form expression $\epsilon^*(x) = \nabla f(x) / \|\nabla f(x)\|$. Here $\rho > 0$ is a hyperparameter controlling the perturbation radius. The algorithm, referred to as *n-SAM*, computes its perturbation using the full-batch gradient. Its update at iteration $k$ is

$$x_{k+1} = x_k - \eta \, \nabla f_{\gamma_k}\Big(x_k + \rho \, \frac{\nabla f(x_k)}{\|\nabla f(x_k)\|}\Big), \tag{3}$$

where $\eta > 0$ is the learning rate, $\rho > 0$ the perturbation radius.

39th Conference on Neural Information Processing Systems (NeurIPS 2025).

Calculating the perturbation on the entire training dataset at each step is prohibitively expensive. Therefore, Foret et al. (2021) suggest estimating the perturbation using a mini-batch, resulting in SAM commonly used in practice. We refer to the practical SAM algorithm as *mini-batch SAM* to distinguish it from other variants. The update rule can be summarized:

$$x_{k+1} = x_k - \eta \nabla f_{\gamma_k} \left( x_k + \rho \frac{\nabla f_{\gamma_k}(x_k)}{\|\nabla f_{\gamma_k}(x_k)\|} \right). \tag{4}$$

Interestingly, it has been observed that although mini-batch SAM was proposed as a computationally efficient variant, it exhibits remarkable generalization ability. In contrast, the original n-SAM (3) offers little to no improvement in generalization (Foret et al., 2021; Andriushchenko and Flammarion, 2022).

**m-SAM and m-sharpness.** *m-SAM* refers to dividing a mini-batch of data into disjoint micro-batches of size $m$, and independently computing perturbations and gradients for each micro-batch, which are then combined to update the parameters. It has been widely observed that the practical performance of m-SAM improves monotonically as $m$ decreases, a phenomenon known as *m-sharpness* (Foret et al., 2021; Behdin et al., 2022; Andriushchenko and Flammarion, 2022). It is worth noting that when $m$ is smaller than the batch size, the perturbation must be computed sequentially across micro-batches, which cannot be parallelized and therefore introduces substantial additional computational overhead, although smaller $m$ typically leads to improved generalization performance.

The update rule for m-SAM can be written as:

$$x_{k+1} = x_k - \frac{\eta m}{|\gamma|} \sum_{\mathcal{I}_j \subset \gamma_k, \ |\mathcal{I}_j| = m} \nabla f_{\mathcal{I}_j} \left( x_k + \rho \frac{\nabla f_{\mathcal{I}_j}(x_k)}{\|\nabla f_{\mathcal{I}_j}(x_k)\|} \right), \tag{5}$$

where $\mathcal{I}_j$ are disjoint subsets of $\gamma_k$, each with size $m$.

In synchronous data-parallel (multi-GPU) training with $D$ devices and per-device batch size $b$, SAM is typically implemented by computing the perturbation *locally* on each device using its own data and then aggregating the perturbed gradients across devices; this implementation is *exactly* an instance of m-SAM with $m = b$. The local-perturbation design avoids an additional cross-device synchronization in the inner perturbation step (only the outer gradient aggregation requires all-reduce), thereby reducing per-step communication overhead. The empirical observation of m-sharpness has thus become a practical cornerstone for deploying SAM at scale.

To provide a theoretical explanation for m-sharpness, we extend the recent Stochastic Differential Equation (SDE) framework of Li et al. (2019); Compagnoni et al. (2023); Luo et al. (2025) by jointly tracking both $\eta$ and $\rho$ to arbitrary expansion orders, providing a unified basis for analyzing SAM and its variants. Under this framework, we derive closed-form drift terms for three unnormalized SAM (USAM) variants, including n-USAM, mini-batch USAM, and m-USAM, revealing how stochastic gradient noise (SGN) drives implicit sharpness regularization and closely correlates with generalization performance. We further extend our analysis to the three normalized (vanilla) SAM variants and observe a similar noise-induced pattern, albeit without closed-form solutions. Motivated by our theory, we propose a sample-reweighting method that uses the magnitude of the SGN as an importance measure.

Our contributions are threefold:

- We develop an extended SDE framework that simultaneously tracks both $\eta$ and $\rho$ to arbitrary orders, and use it to derive SDEs with controllable error terms for n-USAM/SAM, mini-batch USAM/SAM, and m-USAM/SAM.

- We provide a theoretical explanation of the m-sharpness phenomenon, showing how SGN induces an implicit variance-regularization term in the drift, whose strength increases as $m$ decreases. We further present empirical evidence demonstrating that this effect is strongly correlated with generalization performance.

- We introduce *Reweighted SAM (RW-SAM)*, an adaptive reweighting mechanism that assigns larger weights to samples with higher SGN magnitudes, thereby strengthening implicit sharpness regularization; its superior generalization is confirmed through comprehensive experiments.

## 2 Related Work

**Sharpness-Aware Minimization.** Sharpness-Aware Minimization (Foret et al., 2021) has attracted increasing attention due to its consistent improvements in generalization across a wide range of tasks. A growing body of work has been devoted to analyzing and enhancing SAM, including investigations into its generalization principles (Andriushchenko and Flammarion, 2022; Möllenhoff and Khan, 2022; Wen et al., 2022, 2023; Agarwala and Dauphin, 2023; Springer et al., 2024; Luo et al., 2025) and convergence properties (Khanh et al., 2024; Oikonomou and Loizou, 2025), exploring its applications in various domains, and developing algorithmic variants to further improve both generalization (Kwon et al., 2021; Zhuang et al., 2022; Liu et al., 2022b; Kim et al., 2022; Nguyen et al., 2023a,b; Li et al., 2024c; Wu et al., 2024; Tahmasebi et al., 2024; Truong et al., 2024; Li et al., 2024b, 2025; Phan et al., 2025) and computational efficiency (Du et al., 2021; Liu et al., 2022a; Du et al., 2022; Mordido et al., 2023; Tan et al., 2024; Xie et al., 2024).

**m-sharpness.** m-sharpness has long been a mysterious phenomenon in the field of SAM-related research and was first introduced in the original work of Foret et al. (2021). They observed that although SAM theoretically aims to minimize the perturbed loss over the entire training set, its computationally efficient variant, mini-batch SAM, which computes perturbations only at the mini-batch level, outperforms n-SAM, which applies perturbations at the full-batch level. More generally, the generalization performance of m-SAM improves monotonically as $m$ decreases. This phenomenon was further confirmed through extensive experiments in a single-GPU setting by Andriushchenko and Flammarion (2022), and in a multi-GPU setting by Behdin et al. (2022). Although Andriushchenko and Flammarion (2022) proposed several hypotheses to explain it, they were later invalidated by their own experiments, and the underlying cause of this phenomenon remains an open question. It is important to note that our definition of m-sharpness follows the original work of Foret et al. (2021) and the pioneering contributions of Andriushchenko and Flammarion (2022). Some studies have adopted different definitions. For example, Wen et al. (2022) refer to the deterministic algorithm as n-SAM and SAM with a batch size of $1$ as 1-SAM, whereas Behdin et al. (2022) define m as the number of divided micro-batches.

**Structure of stochastic gradient noise.** In expectation, a widely accepted assumption is that the stochastic gradient serves as an unbiased estimator of the full-batch gradient (Jastrzebski et al., 2017; Zhu et al., 2018; HaoChen et al., 2021; Ziyin et al., 2021). Regarding the covariance of SGN, Simsekli et al. (2019) assumed it to be isotropic. However, this view was later challenged by Xie et al. (2020) and Li et al. (2021), who argued that Simsekli et al. (2019) were actually analyzing gradient noise across different iterations rather than noise arising from mini-batch sampling. They provided extensive evidence supporting the idea that the latter can be well modeled as a multivariate Gaussian variable with an anisotropic/parameter-dependent covariance structure. Furthermore, Xie et al. (2023) conducted statistical tests on the Gaussianity to support this perspective.

## 3 Theory

### 3.1 Notation and assumption

In this paper, we denote by $\|\cdot\|$ the Euclidean norm, and the expectation operator $\mathbb{E}$ is taken with respect to the random index set unless otherwise stated. We assume that the stochastic gradient is an unbiased estimator of the full gradient and possesses a finite second moment.

**Assumption 3.1.** We assume that sampling an index $i$ uniformly at random yields i.i.d. stochastic gradients
$$\nabla f_i(x) = \nabla f(x) + \xi_i(x),$$
where $\nabla f(x) := \frac{1}{n} \sum_{i=1}^{n} \nabla f_i(x)$ is the full gradient and $\xi_i(x)$ denotes the SGN. We further assume
$$\mathbb{E}[\xi_i(x)] = 0, \quad \mathrm{Cov}(\xi_i(x)) = V(x),$$
where
$$V(x) := \frac{1}{n} \sum_{i=1}^{n} \nabla f_i(x) \nabla f_i(x)^\top - \nabla f(x) \nabla f(x)^\top.$$

An important consequence of this assumption is $\mathbb{E}\|\nabla f_i(x)\|^2 = \|\nabla f(x)\|^2 + \mathrm{tr}(V(x))$, which we will use repeatedly.

## 3.2 Overview of two-parameter approximation

We extend the existing SDE framework for SAM (Compagnoni et al., 2023; Luo et al., 2025), which can only track $\eta$ to order 1, while ours can jointly track two parameters $\eta$ and $\rho$ to arbitrary expansion orders, thereby decoupling their convergence rates and enabling precise control of the overall approximation error. Specifically, $\eta$ governs the higher-order terms in Dynkin's formula, while $\rho$ captures the remainder term arising from the Taylor expansion (see the detailed formulations in Appendix A). By employing a Dynkin expansion instead of a full Itô–Taylor expansion, it avoids the proliferation of terms and provides a streamlined approach to controlling the remainder error in the two-parameter setting. Another major advantage of this approach is that it allows us to let $\eta$ and $\rho$ tend to zero at independent rates, rather than being constrained to a fixed ratio as in the work of Compagnoni et al. (2023) and Luo et al. (2025). The definition of a two-parameter weak approximation of order $(\alpha, \beta)$ is as follows.

**Definition 3.2** (Two-parameter weak approximation)**.** Let $T > 0$, $0 < \eta < 1$, $0 < \rho < 1$ and set $N = \lfloor T/\eta \rfloor$. Let

$$\{x_k\}_{k=0}^N \quad \text{and} \quad \{X_t\}_{t \in [0,T]}$$

be a discrete-time and a continuous-time stochastic process, respectively. We say that $X_t$ is an *order-$(\alpha, \beta)$ weak approximation* of $x_k$ if, for every $g \in G^{\alpha+1}$, there exists a constant $C > 0$, independent of $\eta, \rho$, such that

$$\max_{0 \le k \le N} \left| \mathbb{E}\big[g(X_{k\eta})\big] - \mathbb{E}\big[g(x_k)\big] \right| \le C\big(\eta^\alpha + \rho^{\beta+1}\big).$$

## 3.3 SDE approximation for USAM variants

We begin by considering USAM (Andriushchenko and Flammarion, 2022; Compagnoni et al., 2023; Dai et al., 2024; Zhou et al., 2024), which is widely studied as a theoretically friendly variant of SAM and can achieve comparable performance to SAM in practice. The update rules for the different variants of the USAM algorithm are shown below. We will see that for USAM, all expressions in the drift term are in closed form, providing an intuitive understanding. In Section 3.4, we will extend our conclusions to the standard SAM.

$$\text{mini-batch USAM:} \quad x_{k+1} = x_k - \eta \, \nabla f_{\gamma_k}\big(x_k + \rho \, \nabla f_{\gamma_k}(x_k)\big) \tag{6}$$

$$\text{n-USAM:} \quad x_{k+1} = x_k - \eta \, \nabla f_{\gamma_k}\big(x_k + \rho \, \nabla f(x_k)\big) \tag{7}$$

$$\text{m-USAM:} \quad x_{k+1} = x_k - \frac{\eta \, m}{|\gamma|} \sum_{\mathcal{I}_j \subset \gamma_k, |\mathcal{I}_j| = m} \nabla f_{\mathcal{I}_j}\big(x_k + \rho \, \nabla f_{\mathcal{I}_j}(x_k)\big) \tag{8}$$

**Theorem 3.3** (Mini-batch USAM SDE - informal statement of Theorem B.2, adapted from Theorem 3.2 of Compagnoni et al. (2023))**.** *Under Assumption 3.1 and mild regularity conditions, the solution of the following SDE (9) is an order-$(1, 1)$ weak approximation of the discrete update of mini-batch USAM (6) with batch size $|\gamma|$:*

$$dX_t = -\nabla\bigg( f(X_t) + \underbrace{\frac{\rho}{2}\|\nabla f(X_t)\|^2 + \frac{\rho}{2|\gamma|}\mathrm{tr}(V(X_t))}_{\textit{implicit regularization}} \bigg)dt + \sqrt{\eta \Sigma^{USAM}(X_t)}dW_t. \tag{9}$$

Based on Eq. (9), it can be observed that under our Assumption 3.1, the *implicit regularization* term of mini-batch USAM (6) can be divided into the gradient of two parts: the squared norm of the full-batch gradient and the trace of the SGN covariance. The latter, which arises additionally from Theorem 3.2 (Compagnoni et al., 2023), is due to the use of random batches for perturbation in mini-batch USAM. We will see that if we use a deterministic perturbation, the regularization effect on the SGN covariance will disappear, as stated in the following theorem:

**Theorem 3.4** (n-USAM SDE - informal statement of Theorem B.4)**.** *Under Assumption 3.1 and mild regularity conditions, the solution of the following SDE (10) is an order-$(1, 1)$ weak approximation of the discrete update of n-USAM (7) with batch size $|\gamma|$:*

$$dX_t = -\nabla\big(f(X_t) + \frac{\rho}{2}\|\nabla f(X_t)\|^2\big)dt + \sqrt{\eta \Sigma^{n-USAM}(X_t)}dW_t. \tag{10}$$

As observed in Table 7 in Appendix J, USAM and SAM exhibit similar behavior under full-batch perturbations, meaning that n-USAM cannot improve generalization performance like mini-batch USAM. Therefore, we argue that the *sharpness regularization* effect of mini-batch USAM actually stems from the last term of its drift term, which is the gradient of the trace of the SGN covariance, i.e., $\nabla \text{tr}(V(X_t))$.

Having understood that n-USAM lacks the sharpness regularization benefits brought by SGN in the drift term, we analyze another contrasting variant, m-USAM, which uses smaller micro-batches to compute the perturbation. Within the framework of the SDE approximation, we can derive the following theorem.

**Theorem 3.5** (m-USAM SDE - informal statement of Theorem B.7). *Under Assumption 3.1 and mild regularity conditions, the solution of the following SDE* (11) *is an order-*$(1,1)$* weak approximation of the discrete update of m-USAM* (8)*:*

$$dX_t = -\nabla\big(f(X_t) + \frac{\rho}{2}\|\nabla f(X_t)\|^2 + \frac{\rho}{2m}\text{tr}(V(X_t))\big)dt + \sqrt{\frac{m\eta}{|\gamma|}\Sigma^{m-USAM}(X_t)}dW_t. \quad (11)$$

It is worth noting that Theorem 3.3 is recovered by setting $m = |\gamma|$. From this SDE approximation, we see that m-USAM's advantages arise from two sources. First, it amplifies the sharpness regularization in the drift term: the coefficient on the SGN covariance changes from $\rho/(2|\gamma|)$ to $\rho/(2m)$ (with $m < |\gamma|$). Second, the diffusion term in m-USAM is reduced by a factor of $m/|\gamma|$ compared to mini-batch USAM with batch size $m$. Since the diffusion term captures the random fluctuations that counteract implicit regularization, shrinking it enhances the stability of the method.

## 3.4 SDE approximation for SAM variants

We now turn to the analysis of (normalized) SAM. With the addition of the normalization factor, the situation becomes much more complex because the expectation of the (unsquared) norm does not have an elementary expression. However, the overall pattern remains similar to the case of USAM. We first present the SDE approximation theorem for SAM variants, then highlight their key differences from the unnormalized version.

**Theorem 3.6** (Mini-batch SAM SDE - informal statement of Theorem C.2, adapted from Theorem 3.5 of Compagnoni et al. (2023)). *Under Assumption 3.1 and mild regularity conditions, the solution of the following SDE* (12) *is an order-*$(1,1)$* weak approximation of the discrete update of mini-batch SAM* (4) *with batch size* $|\gamma|$*:*

$$dX_t = -\nabla\big(f(X_t) + \frac{\rho}{|\gamma|}\mathbb{E}\|\sum_{i\in\gamma}\nabla f_i(X_t)\|\big)dt + \sqrt{\eta\Sigma^{SAM}(X_t)}dW_t. \quad (12)$$

**Theorem 3.7** (n-SAM SDE - informal statement of Theorem C.4). *Under Assumption 3.1 and mild regularity conditions, the solution of the following SDE* (13) *is an order-*$(1,1)$* weak approximation of the discrete update of n-SAM* (3) *with batch size* $|\gamma|$*:*

$$dX_t = -\nabla\big(f(X_t) + \rho\|\nabla f(X_t)\|\big)dt + \sqrt{\eta\Sigma^{n-SAM}(X_t)}dW_t. \quad (13)$$

**Theorem 3.8** (m-SAM SDE - informal statement of Theorem C.7). *Under Assumption 3.1 and mild regularity conditions, the solution of the following SDE* (14) *is an order-*$(1,1)$* weak approximation of the discrete update of m-SAM* (5)*:*

$$dX_t = -\nabla\big(f(X_t) + \frac{\rho}{m}\mathbb{E}\|\sum_{i\in\mathcal{I},|\mathcal{I}|=m}\nabla f_i(X_t)\|\big)dt + \sqrt{\frac{m\eta}{|\gamma|}\Sigma^{m-SAM}(X_t)}dW_t. \quad (14)$$

Unlike the unnormalized algorithm, these SAM regularization terms cannot be expressed as simple functions of the full gradient and SGN covariance. Nonetheless, the following proposition clarifies how the regularization terms in the SDEs depend on the mini/micro batch size, revealing an inverse relationship between the norm and the batch size.

**Proposition 3.9.** *Under Assumption 3.1, the following inequalities hold:*

$$\|\nabla f(x)\| \leq \mathbb{E}\|\nabla f_\gamma(x)\| \leq \sqrt{\|\nabla f(x)\|^2 + \mathbb{E}\|\xi_\gamma(x)\|^2} = \sqrt{\|\nabla f(x)\|^2 + \frac{\text{tr}(V(x))}{|\gamma|}},$$

*where $\xi_\gamma$ denotes the SGN in $\gamma$. Furthermore, if $\nabla f_i(x)$ follows a log-concave distribution, then we have $\mathbb{E}\|\nabla f_\gamma(x)\|$ monotonically increases as $|\gamma|$ decreases.*

A complete proof of Proposition 3.9 is given in Appendix E. Notably, log-concave distributions include many of the standard distributions of interest, such as the Gaussian and exponential distributions.

Based on the above theorems and propositions, we observe that mini-batch SAM (like USAM) regularizes the magnitude of the SGN. In contrast, n-SAM loses this noise-regularization effect, leading to degraded performance, whereas m-SAM amplifies it and thus achieves improved results. One intuitive explanation is that in larger batches, these noise terms tend to cancel each other out, causing the averaged stochastic gradient to concentrate around its expectation and diminishing the contribution of SGN.

### 3.5 Benefits of SGN Covariance Regularization for Generalization

Building on our observation that USAM/SAM variants impose different strengths of sharpness regularization, i.e., variance-based regularization of the SGN, we investigate how directly regularizing the covariance of the SGN further enhances generalization performance.

We address this from two perspectives. First, following the work of Neu et al. (2021); Wang and Mao (2021), a bound on the generalization error can be decomposed into a sum of mutual-information terms, among which the "trajectory term" is controlled by the covariance of the SGN. This implies that by implicitly regularizing the covariance of the SGN during training, one can reduce the cumulative trajectory term over time, thereby tightening the overall generalization bound.

On the other hand, we examine the algorithm's dynamics as it approaches convergence in the late stages of training. As the parameters approach the minimum, the loss and the full gradient on the training set diminish, causing the gradient of the SGN covariance to dominate the drift term. Near a local minimum, it is well established (Jastrzebski et al., 2017; Daneshmand et al., 2018; Zhu et al., 2018; Xie et al., 2020, 2023) that for the negative log-likelihood loss we have

$$V(x) \approx \frac{1}{n} \sum_{i=1}^{n} \nabla f_i(x) \nabla f_i(x)^\top \approx \text{FIM}(x) \approx \nabla^2 f(x), \tag{15}$$

where $\text{FIM}(x)$ denotes the empirical Fisher information matrix and $\nabla^2 f(x)$ the Hessian matrix. Moreover, an empirical result by Xie et al. (2020) shows that for neural networks, this relationship remains approximately valid even far away from the local minima. Therefore, in the late stages of training, regularizing the trace of the SGN covariance is approximately equivalent to regularizing the trace of the Hessian, which is widely regarded as a good measure of sharpness that has been observed to correlate strongly with generalization performance (Keskar et al., 2017; Blanc et al., 2020; Wen et al., 2022; Arora et al., 2022; Damian et al., 2022; Ahn et al., 2023; Tahmasebi et al., 2024).

To verify the dynamics around the minima, we consider the setting from the work of Liu et al. (2020); Damian et al. (2021), where initialization is performed at a bad minimum. We use the checkpoint provided by Damian et al. (2021). The learning rate is set to $1\mathrm{e}{-3}$, under which SGD has been shown to struggle to escape poor minima. We do not employ any explicit regularization techniques. For SAM, we set $\rho = 5\mathrm{e}{-3}$ and compare the performance of m-SAM with different values of $m$. In Figures 1 and 2, we can clearly observe that as $m$ decreases, the regularization effect on the SGN covariance is significantly strengthened, which in turn accelerates the escape from poor minima—consistent with our theoretical analysis.

## 4 Practical Method

In this section, we will demonstrate how to translate the theoretical insights gained from our SDE approximations into a practical method. While m-SAM[1] achieves substantially better generalization than mini-batch SAM by strengthening the regularization of the SGN magnitude, its inherently sequential nature creates a serious parallelization bottleneck (see Table 7 in Appendix J for performance and time cost). This is because it must sequentially compute the perturbation for each micro-batch

---

[1]In the multi-GPU setting, we use "m-SAM" to denote the scenario where $m <$ per-device batch size, which is inherently not parallelizable as well.

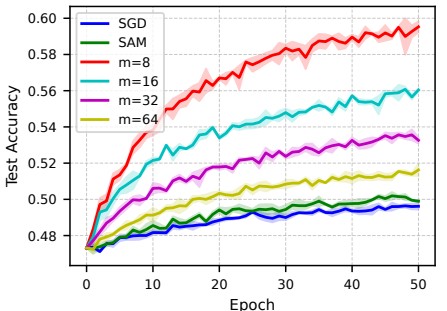
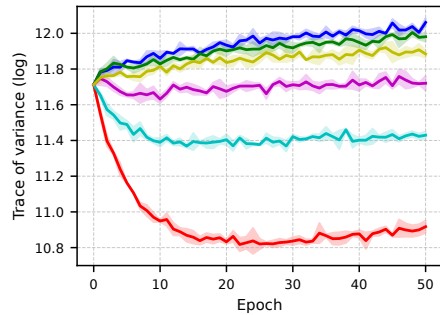

Figure 1: Speed of escaping poor minima, measured by test accuracy.

Figure 2: Variance of SGN over iterations.

and then individually backpropagate to obtain the perturbed gradient. Building on our theoretical insights, a natural question arises:

*Can we design a parallelizable algorithm that preserves the generalization advantages of m-SAM?* One important observation is, if we further assume that the SGN is approximately orthogonal to the full gradient, or that the norm of the full gradient is negligible compared to that of the SGN (as often occurs in the late stages of training when the model approaches convergence), which commonly arises in signal-to-noise-ratio analyses (Cao et al., 2022; Jelassi et al., 2022; Zou et al., 2023; Huang et al., 2023; Allen-Zhu and Li, 2023; Han et al., 2024; Huang et al., 2024; Li et al., 2024a), then from the following equation we observe that the norm of the stochastic gradient is directly related to the norm of the SGN:

$$\|\nabla f_i(x)\| \approx \sqrt{\|\nabla f(x)\|^2 + \|\xi_i\|^2}. \tag{16}$$

This decomposition reveals that samples with larger gradient norms carry higher-magnitude SGN. Drawing inspiration from importance sampling, which assigns a weight to each sample proportional to its "importance", we adapt this idea to emphasize more important samples when computing SAM's perturbation (2). In our framework, we quantify importance by the magnitude of the SGN: samples exhibiting larger SGN contribute more to sharpness regularization and therefore deserve greater weight in the perturbation. Specifically, we introduce an *adaptive weighting mechanism* in which each weight $p_i$ reflects the importance of sample $i$ when computing the perturbation vector. This reweighting strategy defines a probability distribution $P = \{p_i\}_{i \in \gamma}$ over the sampled indices $i \in \gamma$, thereby modulating the influence of each sample. We refer to the resulting algorithm as *Reweighted SAM (RW-SAM)*. The objective of RW-SAM's perturbation is formulated as follows:

$$\max_{P \in \Delta} \quad \max_{\|\epsilon\| \leq 1} \quad \Big\langle \sum_{i \in \gamma} p_i \, \nabla f_i(x), \epsilon \Big\rangle + \frac{\mathbb{H}(P)}{\lambda}, \tag{17}$$

where $\Delta$ denotes the probability simplex, $\mathbb{H}$ is the entropy function used to prevent all weights from concentrating on a single sample, and $\lambda$ is a hyperparameter to maintain a balance between emphasis and diversity. Notably, as $\lambda \to 0$, Objective (17) degenerates to mini-batch SAM's perturbation (2).

Solving Objective (17) for $\epsilon$ yields:

$$\epsilon^* = \frac{\sum_{i \in \gamma} p_i \, \nabla f_i(x)}{\left\| \sum_{i \in \gamma} p_i \, \nabla f_i(x) \right\|}. \tag{18}$$

Since Objective (17) does not have a closed-form solution for $P$, we propose solving its relaxation:

$$\max_{P \in \Delta} \quad \sum_{i \in \gamma} p_i \big\| \nabla f_i(x) \big\| + \frac{\mathbb{H}(P)}{\lambda}. \tag{19}$$

Objective (19) corresponds to the well-known Gibbs distribution (see derivation in Section H), which is given by:

$$p_i^* = \frac{\exp\big(\lambda \left\| \nabla f_i(x) \right\|\big)}{\sum_{j \in \gamma} \exp\big(\lambda \left\| \nabla f_j(x) \right\|\big)}. \tag{20}$$

Broadly speaking, Eq. (20) corresponds to assigning higher weights to samples with larger stochastic gradient norms, where $\lambda$ controls the concentration of this distribution. In practice, the optimal value of $\lambda$ depends on the scale of per-sample gradient norms. Therefore, we normalize the estimated gradient norms before applying the exponential function, making the algorithm's performance less sensitive to the choice of $\lambda$. As observed in Section 5.4, normalization significantly reduces the need for tuning $\lambda$.

Additionally, we propose using a finite-difference method combined with Monte Carlo sampling to avoid per-sample gradient-norm estimation via backpropagation. The formula is as follows (see derivation in Appendix G):

$$\|\nabla f_i(x)\| \approx \sqrt{\frac{1}{Q} \sum_{q=1}^{Q} \left( \frac{f_i(x + \delta z_q) - f_i(x)}{\delta} \right)^2},$$
(21)

where $z \in \mathbb{R}^d$ is a Rademacher random vector, $\delta$ is a small constant. This estimator requires only $Q$ additional forward passes, as we can conveniently obtain the loss for each sample in a single forward pass, making it an efficient way to estimate the gradient norm for each sample. To minimize additional computational overhead, we set $Q = 1$ in our experiments, consistent with common practice in deep learning (Kingma et al., 2013; Gal and Ghahramani, 2016; Ho et al., 2020; Malladi et al., 2023), and found that this choice suffices to achieve a non-trivial performance improvement. However, unlike common implementations (Kingma et al., 2013; Ho et al., 2020; Malladi et al., 2023), we propose using Rademacher instead of Gaussian perturbations to minimize the variance of the Monte Carlo estimator. Since Rademacher variables have a fixed expected squared norm, they achieve the optimal variance (according to Theorem 2.2 in Ma and Huang (2025)). The pseudocode for our algorithm is presented in Algorithm 1 (see Appendix I).

## 5 Experiments

### 5.1 Training from scratch

We evaluate three optimization methods: SGD, SAM[2], and RW-SAM on CIFAR-10 and CIFAR-100 (Krizhevsky et al., 2009), training three models from scratch: ResNet-18, ResNet-50 (He et al., 2016), and WideResNet-28-10 (Zagoruyko and Komodakis, 2016). We use a batch size of 128 and a cosine learning rate schedule with an initial learning rate of 0.1. SAM and RW-SAM are trained for 200 epochs, while SGD is trained for 400 epochs. We apply a momentum of 0.9 and a weight decay of $5e-4$, along with standard data augmentation techniques, including horizontal flipping, padding by four pixels, and random cropping.

For SAM and RW-SAM, we set $\rho = 0.05$ for CIFAR-10 and $\rho = 0.1$ for CIFAR-100. In the case of RW-SAM, we determine $\delta$ through finite-difference estimation on the model before training, based on the estimated error. Specifically, we consider $\delta \in \{1e-5, 1e-4, 1e-3\}$ and use $\delta = 1e-3$ for ResNet-18 and $\delta = 1e-4$ for both ResNet-50 and WideResNet-28-10. For the additional hyperparameter $\lambda$ in RW-SAM, we performed a grid search over $\{0.25, 0.5, 1.0, 2.0\}$ on a validation set and found that 0.5 consistently yielded strong performance across experiments.

Table 1: Test accuracy comparison on CIFAR-10 and CIFAR-100 with different optimizers.

| Model | CIFAR-10 | | | CIFAR-100 | | |
|---|---|---|---|---|---|---|
| | SGD | SAM | RW-SAM | SGD | SAM | RW-SAM |
| ResNet-18 | $95.62 \pm 0.03$ | $95.99 \pm 0.07$ | $\mathbf{96.24 \pm 0.05}$ | $78.91 \pm 0.18$ | $78.90 \pm 0.27$ | $\mathbf{79.31 \pm 0.28}$ |
| ResNet-50 | $95.64 \pm 0.37$ | $96.06 \pm 0.04$ | $\mathbf{96.34 \pm 0.04}$ | $79.55 \pm 0.16$ | $80.31 \pm 0.35$ | $\mathbf{80.83 \pm 0.05}$ |
| WideResNet | $96.47 \pm 0.03$ | $96.91 \pm 0.02$ | $\mathbf{97.11 \pm 0.05}$ | $81.55 \pm 0.15$ | $83.25 \pm 0.07$ | $\mathbf{83.52 \pm 0.08}$ |

For large-scale experiments, we train a ResNet-50 on ImageNet-1K (Deng et al., 2009) for 90 epochs with an initial learning rate of 0.05. For both SAM and RW-SAM, we use the same $\rho = 0.05$. For the additional hyperparameter $\lambda$ in RW-SAM, we set $\lambda = 0.25$. All other hyperparameters remain the same as those used on CIFAR-10/100.

---

[2]In this section, consistent with common practice, we use "SAM" to refer to the mini-batch SAM.

We repeated three independent experiments and reported the mean and standard deviation of the test accuracy in Table 1 and Table 2a. We observe that RW-SAM consistently outperforms the baselines across various models, as well as on both small and large datasets.

**Analysis of Computational Overhead.** RW-SAM requires an additional forward pass to estimate per-sample gradient norms, leading to approximately 1/6 more training overhead compared to vanilla SAM, as a forward pass typically takes about half the time of a backward pass (Kaplan, 2022). For a report of the additional wall-clock time overhead, please see Table 8 in Appendix J. However, RW-SAM matches the performance of m-SAM at $m = 64$ without incurring its nearly two-fold training overhead (See Table 7 in Appendix J). This highlights the efficiency of RW-SAM in balancing computational cost and performance.

Table 2: (a) Test accuracy on ImageNet-1K with different optimizers; (b) Test accuracy fine-tuning ViT-B/16 on CIFAR-10/100 with different optimizers.

| ImageNet-1K | SGD | SAM | RW-SAM |
|---|---|---|---|
| ResNet-50 | $76.67 \pm 0.05$ | $77.16 \pm 0.04$ | $\mathbf{77.37 \pm 0.05}$ |

(a)

| | SGD | SAM | RW-SAM |
|---|---|---|---|
| CIFAR-10 | $98.24 \pm 0.05$ | $98.40 \pm 0.02$ | $\mathbf{98.58 \pm 0.02}$ |
| CIFAR-100 | $88.71 \pm 0.10$ | $89.63 \pm 0.12$ | $\mathbf{89.89 \pm 0.09}$ |

(b)

## 5.2 Fine-tuning

We fine-tune a ViT-B/16 model (Dosovitskiy et al., 2020), pre-trained on ImageNet-1K, on CIFAR-10 and CIFAR-100. We train for 20 epochs with an initial learning rate of 0.01. Other hyperparameters are the same as those in training from scratch. The results are summarized in Table 2b. We also fine-tune a pretrained DistilBERT model (Sanh et al., 2019) on the GLUE benchmark (Wang et al., 2018). We use AdamW (Loshchilov and Hutter, 2019) as the base optimizer. The detailed hyperparameter settings are provided in Table 10 of Appendix J, and the results are presented in Table 3. We observe that RW-SAM consistently outperforms SAM in both fine-tuning experiments.

Table 3: Performance comparison on GLUE tasks using different optimizers.

| Optimizer | CoLA | MNLI | MRPC | QNLI | QQP | RTE | SST-2 | STS-B | Average |
|---|---|---|---|---|---|---|---|---|---|
| AdamW | 0.538 | 0.825 | 0.900 | 0.884 | 0.868 | **0.628** | 0.914 | 0.864 | 0.804 |
| SAM | 0.517 | 0.824 | 0.900 | 0.894 | 0.871 | 0.610 | 0.911 | **0.870** | 0.800 |
| RW-SAM | **0.560** | **0.826** | **0.904** | **0.896** | **0.872** | 0.625 | **0.915** | **0.870** | **0.809** |

## 5.3 Robustness to label noise

Foret et al. (2021) have shown that SAM exhibits robustness to label noise. Motivated by this, we evaluate RW-SAM's performance on CIFAR-10 with labels randomly flipped at specified noise ratios. We train a ResNet-18 and report clean test accuracies in Table 4. As the noise ratio increases, RW-SAM maintains remarkably strong performance. In particular, at an 80% noise ratio, RW-SAM achieves a 16% absolute accuracy improvement over SAM.

## 5.4 Additional experiments

**Hyperparameter sensitivity.** We train ResNet-18 on CIFAR-100 using RW-SAM to evaluate its sensitivity to different values of $\lambda$. As summarized in Table 5, RW-SAM demonstrates robustness to the choice of $\lambda$ and consistently outperforms the baselines within a reasonable range.

**Trace of stochastic gradient covariance.** According to our theory, we compare the trace of the stochastic gradient covariance matrix at convergence for different algorithms, which, near the minimum, closely approximates the Hessian matrix (See Eq. (15)). The results, shown in Table 6, indicate that compared to SGD and SAM, RW-SAM indeed converges to a minimum with a smaller stochastic gradient covariance magnitude.

Table 4: Performance comparison on CIFAR-10 across different noise ratios

| Noise Ratio | SGD | SAM | RW-SAM |
|---|---|---|---|
| 20% | $87.54 \pm 0.20$ | $90.01 \pm 0.09$ | $\mathbf{90.34} \pm 0.20$ |
| 40% | $83.66 \pm 0.30$ | $86.40 \pm 0.12$ | $\mathbf{86.87} \pm 0.11$ |
| 60% | $76.64 \pm 0.32$ | $78.79 \pm 0.24$ | $\mathbf{81.52} \pm 0.31$ |
| 80% | $46.53 \pm 1.13$ | $37.69 \pm 3.12$ | $\mathbf{53.17} \pm 3.70$ |

Table 5: RW-SAM $\lambda$-sensitivity

| $\lambda$ | 0.25 | 0.5 | 1.0 | 2.0 |
|---|---|---|---|---|
| | $79.09 \pm 0.21$ | $\mathbf{79.31} \pm 0.28$ | $79.12 \pm 0.33$ | $79.03 \pm 0.22$ |

Table 6: Trace of the gradient covariance

| Optimizer | Trace |
|---|---|
| SGD | $572.39 \pm 24.15$ |
| SAM | $198.40 \pm 6.20$ |
| RW-SAM | $\mathbf{177.79} \pm 5.10$ |

# 6   Conclusion

In this work, we conducted a comprehensive theoretical analysis of SAM and its variants through an enhanced SDE modeling framework. Our findings reveal that the structure of SGN plays a crucial role in implicit regularization, significantly influencing generalization performance. Based on our analysis, we proposed Reweighted SAM, an adaptive weighting mechanism for perturbation, which we empirically validated through extensive experiments. Our study provides a deeper understanding of the dynamics of SAM-based algorithms and offers new perspectives on improving their generalization performance.

## Acknowledgement

Trung Le, Mehrtash Harandi, and Dinh Phung were supported by the ARC Discovery Project grants DP230101176 and DP250100262. Trung Le and Mehrtash Harandi were also supported by the Air Force Office of Scientific Research under award number FA9550-23-S-0001.

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

## A  General theory for two-parameter weak approximation.

Let $T > 0$, $\eta \in (0, \min\{1, T\})$, and $N = \lfloor T/\eta \rfloor$. We consider the general discrete iteration

$$x_{k+1} \;=\; x_k + \eta\, h\big(x_k, \gamma_k, \eta, \rho\big), \qquad x_0 \in \mathbb{R}^d, \quad k = 0, 1, \ldots, N, \tag{22}$$

and its corresponding continuous-time approximation by the SDE

$$\mathrm{d}X_t \;=\; b\big(X_t, \eta, \rho\big)\,\mathrm{d}t \;+\; \sqrt{\eta}\,\sigma\big(X_t, \eta, \rho\big)\,\mathrm{d}W_t, \qquad X_0 = x_0,\ t \in [0, T]. \tag{23}$$

We denote $\widetilde{X}_k := X_{k\eta}$, and the one-step changes:

$$\Delta(x) := x_1 - x, \qquad \widetilde{\Delta}(x) := \widetilde{X}_1 - x. \tag{24}$$

Following the SDE framework by Mil'shtein (1986); Li et al. (2017); Compagnoni et al. (2023); Luo et al. (2025), we have the following definition:

**Definition A.1.** Let $G$ denote the set of continuous functions $\mathbb{R}^d \to \mathbb{R}$ of at most polynomial growth, i.e. $g \in G$ if there exists positive integers $\kappa_1, \kappa_2 > 0$ such that

$$|g(x)| \leq \kappa_1(1 + \|x\|^{2\kappa_2}),$$

for all $x \in \mathbb{R}^d$. Moreover, for each integer $\alpha \geq 1$ we denote by $G^\alpha$ the set of $\alpha$-times continuously differentiable functions $\mathbb{R}^d \to \mathbb{R}$ which, together with its partial derivatives up to and including order $\alpha$, belong to $G$.

This definition comes from the field of numerical analysis of SDEs (Mil'shtein, 1986). In the case of $g(x) = \|x\|^j$, the bound restricts the difference between the $j$-th moments of the discrete process and those of the continuous process. We write $\mathcal{O}(\eta^\alpha \rho^\beta)$ to denote that there exists a function $K \in G$ independent of $\rho$, $\eta$, such that the error terms are bounded by $K\eta^\alpha \rho^\beta$.

**Theorem A.2.** *(Adaptation of Theorem 3 in Li et al. (2019)) Let $T > 0$, $\eta \in (0, \min\{1, T\})$, $N = \lfloor T/\eta \rfloor$. Let $\alpha \geq 1$ be an integer. Suppose further that the following conditions hold:*

*(i) There exists $K_1 \in G$, independent of $\eta, \rho$, such that for each $s = 1, 2, \ldots, \alpha$ and any indices $i_1, \ldots, i_s \in \{1, \ldots, d\}$,*

$$\left| \mathbb{E}\prod_{j=1}^{s} \Delta_{(i_j)}(x) \;-\; \mathbb{E}\prod_{j=1}^{s} \widetilde{\Delta}_{(i_j)}(x) \right| \;\leq\; K_1(x)\,(\eta^{\alpha+1} + \eta\rho^{\beta+1}),$$

*and*

$$\mathbb{E}\prod_{j=1}^{\alpha+1} \big|\Delta_{(i_j)}(x)\big| \;\leq\; K_1(x)\,(\eta^{\alpha+1} + \eta\rho^{\beta+1}).$$

*(ii) For each $m \geq 1$, the $2m$-moment of $x_k$ is uniformly bounded in $k$ and $\eta$, i.e. there exists $K_2 \in G$, independent of $\eta$ and $k$, such that*

$$\mathbb{E}\big\|x_k\big\|^{2m} \leq K_2(x), \quad k = 0, 1, \ldots, N.$$

*Then, for each $g \in G^{\alpha+1}$, there exists a constant $C > 0$, independent of $\eta, \rho$, such that*

$$\max_{0 \leq k \leq N} \big| \mathbb{E}\, g(x_k) \;-\; \mathbb{E}\, g(X_{k\eta}) \big| \;\leq\; C\,(\eta^\alpha + \rho^{\beta+1}).$$

By substituting Assumption (i) in the penultimate step of the proof in Theorem 3 in Li et al. (2019) with Assumption (i) of our Theorem A.2, the same argument goes through and yields the desired conclusion. Hence, we omit the proof.

Next, we state the key lemma for the two-parameter weak approximation. By employing a Dynkin expansion (see, e.g. Evans (2012)) instead of a full Itô–Taylor expansion, it avoids the proliferation of terms and provides a streamlined approach to controlling the remainder error in the two-parameter setting.

**Lemma A.3** (Two-parameter Dynkin (semigroup) expansion). *Let $\psi \in G^{2\alpha+2}$, and suppose the drift admits the expansion*

$$b(x, \rho) = \sum_{m=0}^{\beta} \rho^m \, b_m(x) \; + \; O(\rho^{\beta+1}), \qquad \sigma(x) = \sigma_0(x).$$

*Define*

$$A_m \, \psi(x) := b_m^{(i)}(x) \, \partial_i \psi(x) \quad (m = 0, 1, \dots, \beta), \qquad A_\Delta \, \psi(x) := \tfrac{1}{2} \left[\sigma_0 \sigma_0^T\right]^{(ij)} \partial_{ij}^2 \psi(x).$$

*Suppose further that $b_m \ (m = 0, 1, \dots, \beta), \sigma_0 \in G^{2\alpha}$, then for any nonnegative integers $\alpha, \beta$,*

$$\mathbb{E}\big[\psi(X_\eta)\big] = \sum_{n=0}^{\alpha} \frac{\eta^n}{n!} \sum_{m=0}^{\beta} \rho^m \sum_{\substack{\ell \in \{0, \Delta, 1, \dots, \beta\}^n \\ |\ell| = m}} A_{\ell_1} A_{\ell_2} \cdots A_{\ell_n} \psi(x) \; + \; O(\eta^{\alpha+1}) \; + \; O(\eta \rho^{\beta+1}).$$

*Here $|\ell| := \sum_{i: \, \ell_i \notin \{0, \Delta\}} \ell_i$, and each multi-index $\ell = (\ell_1, \dots, \ell_n)$ contributes a factor $\rho^{|\ell|}$, and indices $\ell_i = 0$ or $\ell_i = \Delta$ contribute no $\rho$-power (via $A_0$ or $A_\Delta$).*

*Proof.* Let

$$L \, \phi(x) \; = \; \big(b(x, \rho) \cdot \nabla \phi\big)(x) \; + \; \tfrac{1}{2} [\sigma_0 \sigma_0^T]^{(ij)}(x) \, \partial_{ij}^2 \phi(x)$$

be the infinitesimal generator of the diffusion. Since

$$b(x, \rho) = \sum_{m=0}^{\beta} \rho^m \, b_m(x) + O(\rho^{\beta+1}) \,,$$

we may write

$$L = \sum_{m=0}^{\beta} \rho^m \, A_m \; + \; A_\Delta \; + \; O(\rho^{\beta+1}) \,,$$

where $A_m$ and $A_\Delta$ are as in the statement. By Dynkin's formula (or equivalently the semigroup expansion),

$$\mathbb{E}\big[\psi(X_\eta)\big] = \big(e^{\eta L} \psi\big)(x) = \sum_{n=0}^{\alpha} \frac{\eta^n}{n!} L^n \psi(x) \; + \; O(\eta^{\alpha+1}) \,.$$

It remains to expand each power $L^n$. By the multinomial theorem,

$$L^n = \Big(\sum_{m=0}^{\beta} \rho^m A_m + A_\Delta + O(\rho^{\beta+1})\Big)^n = \sum_{m=0}^{\beta} \rho^m \sum_{\substack{\ell \in \{0, \Delta, 1, \dots, \beta\}^n \\ |\ell| = m}} A_{\ell_1} A_{\ell_2} \cdots A_{\ell_n} \; + \; O(\rho^{\beta+1}) \,.$$

Hence

$$\frac{\eta^n}{n!} L^n \psi(x) = \frac{\eta^n}{n!} \sum_{m=0}^{\beta} \rho^m \sum_{\substack{\ell \in \{0, \Delta, 1, \dots, \beta\}^n \\ |\ell| = m}} A_{\ell_1} A_{\ell_2} \cdots A_{\ell_n} \psi(x) \; + \; O(\eta^{n+1}) + O(\eta \rho^{\beta+1}).$$

Summing over $n = 0, 1, \dots, \alpha$ and collecting the remainders $O(\eta^{\alpha+1})$ and $O(\eta \rho^{\beta+1})$ yields exactly the claimed two-parameter expansion. $\qquad\square$

Since our focus in this paper is on the order-(1,1) weak approximation, we now present the one-step approximation lemma for SDEs in the case $\alpha = \beta = 1$, as follows. For readers interested in higher-order two-parameter weak approximations, it is sufficient to apply higher-order truncations of the Dynkin and Taylor expansions in the two lemmas below and then match the corresponding moments at each order.

**Lemma A.4** (One-step moment estimates up to $\eta^1, \rho^1$ for SDEs)**.** *Suppose the drift admits the expansion*

$$b(x, \rho) = b_0(x) + \rho \, b_1(x) + O(\rho^2), \qquad \sigma(x) = \sigma_0(x),$$

*and assume $b_0, b_1, \sigma_0 \in G^2$. Let $\widetilde{\Delta}(x)$ be the one-step increment defined in (24). Then:*

*(i)* $\mathbb{E}[\widetilde{\Delta}_{(i)}(x)] = \eta\big(b_0^{(i)}(x) + \rho\, b_1^{(i)}(x)\big) + O(\eta^2) + O(\eta\,\rho^2).$

*(ii)* $\mathbb{E}\big[\widetilde{\Delta}_{(i)}(x)\,\widetilde{\Delta}_{(j)}(x)\big] = \eta^2\big(b_0^{(i)}(x)\,b_0^{(j)}(x) + \sum_k \sigma_0^{(i,k)}(x)\,\sigma_0^{(j,k)}(x)\big) + \eta^2\rho\big(b_0^{(i)}(x)\,b_1^{(j)}(x) +$

$b_1^{(i)}(x)\,b_0^{(j)}(x)\big) + O(\eta^2\rho^2) + O(\eta^3).$

*(iii)* $\mathbb{E}\Big[\prod_{j=1}^3 |\widetilde{\Delta}_{(i_j)}(x)|\Big] = O(\eta^3).$

*Proof.* For each $s = 1, 2, 3$ and any choice of indices $i_1, \ldots, i_s$, define the test function

$$\psi_s(z) = \prod_{j=1}^s \big(z_{(i_j)} - x_{(i_j)}\big).$$

Since $\psi_s \in C^4(\mathbb{R}^d)$ with at most polynomial growth, we may invoke Lemma A.2 with truncation orders $\alpha = 1$ and $\beta = 1$. This yields,

$$\mathbb{E}\big[\psi(x)\big] = \psi(x) + \eta \sum_{\ell \in \{0, \Delta\}} A_\ell \psi(x) + \eta\rho\, A_1 \psi(x) + O(\eta^2) + O(\eta\rho^2).$$

(i) *First moment.* Here

$$\psi_1(z) = z_{(i)} - x_{(i)},$$

Hence

$$\mathbb{E}[\widetilde{\Delta}_{(i)}(x)] = \eta\, b_0^{(i)}(x) + \eta\,\rho\, b_1^{(i)}(x) + O(\eta^2) + O(\eta\,\rho^2),$$

proving (i).

(ii) *Second moment.* Now

$$\psi_2(z) = (z_{(i)} - x_{(i)})(z_{(j)} - x_{(j)}),$$

It follows that

$$\mathbb{E}\big[\widetilde{\Delta}_{(i)}(x)\,\widetilde{\Delta}_{(j)}(x)\big] = \eta^2\Big[b_0^{(i)}b_0^{(j)} + \sum_k \sigma_0^{(i,k)}\sigma_0^{(j,k)}\Big] + \eta^2\rho\big[b_0^{(i)}b_1^{(j)} + b_1^{(i)}b_0^{(j)}\big] + O(\eta^2\rho^2) + O(\eta^3),$$

proving (ii).

(iii) *Third moment.* Finally,

$$\psi_3(z) = \prod_{j=1}^3 \big(z_{(i_j)} - x_{(i_j)}\big),$$

and since each nonzero term in the expansion has total order $n \geq 3$, Lemma A.2 gives

$$\mathbb{E}\Big[\prod_{j=1}^3 |\widetilde{\Delta}_{(i_j)}(x)|\Big] = \mathbb{E}\big[|\psi_3(X_{k+1})|\big] = O(\eta^3),$$

establishing (iii). This completes the proof. $\qquad\square$

Similar to the continuous-time setting, we require the following one-step error lemma for the discrete algorithm in the case $\alpha = \beta = 1$:

**Lemma A.5** (One-step moment estimates up to $\eta^1, \rho^1$ for the discrete algorithm)**.** *Suppose the discrete update 22 admits the expansion*

$$h(x, \gamma, \rho) = h_0(x, \gamma) + \rho\, h_1(x, \gamma) + O(\rho^2),$$

*and assume $h_0, h_1 \in G^2$. Let $\Delta(x)$ be the one-step increment defined in (24). Then:*

*(i)* $\mathbb{E}[\Delta_{(i)}(x)] = \eta h_0^{(i)}(x) + \eta\rho\, h_1^{(i)}(x) + O(\eta\,\rho^2).$

*(ii)* $\mathbb{E}\big[\Delta_{(i)}(x)\,\Delta_{(j)}(x)\big] = \eta^2\big(h_0^{(i)}(x)\,h_0^{(j)}(x) + \Sigma_{0,0}^{(ij)}(x)\big) + \eta^2\rho\big(h_0^{(i)}(x)\,h_1^{(j)}(x) +$
$h_1^{(i)}(x)\,h_0^{(j)}(x) + \Sigma_{0,1}^{(ij)}(x) + \Sigma_{1,0}^{(ij)}(x)\big) + O(\eta^2\rho^2).$

*(iii)* $\mathbb{E}\Big[\prod_{j=1}^{3}\big|\Delta_{(i_j)}(x)\big|\Big] = O(\eta^3).$

*where* $h_0(x) = \mathbb{E}h_0(x,\gamma)$, $h_1(x) = \mathbb{E}h_1(x,\gamma)$, $\Sigma_{0,0}(x) = \mathrm{Cov}(h_0(x,\gamma), h_0(x,\gamma))$, $\Sigma_{0,1}(x) = \mathrm{Cov}(h_0(x,\gamma), h_1(x,\gamma))$.

*Proof.* Recall that

$$\Delta(x) \;=\; \eta\, h(x,\gamma,\rho) \;=\; \eta\big(h_0(x,\gamma) + \rho\, h_1(x,\gamma) + O(\rho^2)\big).$$

Hence for each coordinate $i$,

(i) *First moment.*

$$\begin{aligned}
\mathbb{E}[\Delta_{(i)}(x)] &= \eta\, \mathbb{E}\big[h_0^{(i)}(x,\gamma) + \rho\, h_1^{(i)}(x,\gamma) + O(\rho^2)\big]\\
&= \eta\big(h_0^{(i)}(x) + \rho\, h_1^{(i)}(x)\big) + \; O(\eta\,\rho^2).
\end{aligned}$$

(ii) *Second moment.*

$$\begin{aligned}
\mathbb{E}\big[\Delta_{(i)}(x)\,\Delta_{(j)}(x)\big] &= \eta^2\, \mathbb{E}_\gamma\Big[\big(h_0^{(i)}(x,\gamma) + \rho\, h_1^{(i)}(x,\gamma)\big)\big(h_0^{(j)}(x,\gamma) + \rho\, h_1^{(j)}(x,\gamma)\big)\Big] + O(\eta^2\rho^2)\\
&= \eta^2\Big\{\mathbb{E}_\gamma\big[h_0^{(i)}(x,\gamma)\,h_0^{(j)}(x,\gamma)\big] + \rho\,\big(\mathbb{E}_\gamma[h_0^{(i)}(x,\gamma)\,h_1^{(j)}(x,\gamma)]\\
&\quad + \mathbb{E}_\gamma[h_1^{(i)}(x,\gamma)\,h_0^{(j)}(x,\gamma)]\big)\Big\} + O(\eta^2\rho^2)\\
&= \eta^2\Big\{h_0^{(i)}(x)\,h_0^{(j)}(x) + \Sigma_{0,0}^{(ij)}(x)\Big\}\\
&\quad + \eta^2\rho\Big\{h_0^{(i)}(x)\,h_1^{(j)}(x) + h_1^{(i)}(x)\,h_0^{(j)}(x) + \Sigma_{0,1}^{(ij)}(x) + \Sigma_{1,0}^{(ij)}(x)\Big\} + O(\eta^2\rho^2).
\end{aligned}$$

(iii) *Third moment.* Since $h_0, h_1 \in G^2$ implies that all moments up to order three are finite and $\Delta = O(\eta)$, we have

$$\mathbb{E}\Big[\big|\Delta_{(i_1)}\Delta_{(i_2)}\Delta_{(i_3)}\big|\Big] = O(\eta^3).$$

This completes the proof. $\qquad\square$

# B  SDE approximation for USAM variants

Recall that the update rules of USAM variants are defined by:

$$\text{mini-batch USAM:}\quad x_{k+1} = x_k - \eta\,\nabla f_{\gamma_k}\big(x_k + \rho\,\nabla f_{\gamma_k}(x_k)\big) \tag{25}$$

$$\text{n-USAM:}\quad x_{k+1} = x_k - \eta\,\nabla f_{\gamma_k}\big(x_k + \rho\,\nabla f(x_k)\big) \tag{26}$$

$$\text{m-USAM:}\quad x_{k+1} = x_k - \frac{\eta\, m}{|\gamma|}\sum_{\mathcal{I}_j \subset \gamma_k, |\mathcal{I}_j|=m} \nabla f_{\mathcal{I}_j}\big(x_k + \rho\,\nabla f_{\mathcal{I}_j}(x_k)\big) \tag{27}$$

In this section, we impose the following growth assumption on the functions $f$ and $f_\gamma$:

**Assumption B.1.** The functions $f$ and $f_i$ belong to the class $G^4$.

## B.1  Mini-batch USAM

For the mini-batch USAM algorithm (25), we define the continuous-time approximation $X_t$ as the solution to the following SDE:

$$dX_t = -\nabla f^{USAM}(X_t)\,dt + \sqrt{\eta\,\Sigma^{USAM}(X_t)}\,dW_t, \tag{28}$$

where

$$f^{USAM}(X_t) := f(X_t) + \frac{\rho}{2}\|\nabla f(X_t)\|^2 + \frac{\rho}{2|\gamma|}\mathrm{tr}(V(X_t))$$

$$\Sigma^{USAM}(X_t) := \Sigma_{0,0}(X_t) + \rho\big(\Sigma_{0,1}(X_t) + \Sigma_{0,1}^\top(X_t)\big). \tag{29}$$

$$\Sigma_{0,0}(X_t) := \mathbb{E}\left[\big(\nabla f_\gamma(X_t) - \nabla f(X_t)\big)\big(\nabla f_\gamma(X_t) - \nabla f(X_t)\big)^\top\right]$$

$$\Sigma_{0,1}(X_t) := \mathbb{E}\big[\big(\nabla f_\gamma(X_t) - \nabla f(X_t)\big) \cdot \big(\nabla^2 f_\gamma(X_t)\nabla f_\gamma(X_t) - \mathbb{E}[\nabla^2 f_\gamma(X_t)\nabla f_\gamma(X_t)]\big)^\top\big].$$

**Theorem B.2** (mini-batch USAM SDE, adapted from Theorem 3.2 of Compagnoni et al. (2023)). *Under Assumptions 3.1 and B.1, let $0 < \eta < 1$, $T > 0$, and $N = \lfloor T/\eta \rfloor$. Denote by $\{x_k\}_{k=0}^N$ the mini-batch USAM iterates in (6), and let $\{X_t\}_{t\in[0,T]}$ be the solution of the SDE (28). Suppose:*

*(i) The functions*

$$\nabla f^{\text{USAM}} \;=\; \nabla\Big(f + \frac{\rho}{2}\|\nabla f\|^2 + \frac{\rho}{2|\gamma|}\text{tr}(V)\Big) \quad \text{and} \quad \sqrt{\Sigma^{\text{USAM}}}$$

*are Lipschitz on $\mathbb{R}^d$.*

*(ii) The mapping*

$$h_\gamma(x) \;=\; -\nabla f_\gamma\big(x + \rho\,\nabla f_\gamma(x)\big)$$

*satisfies, almost surely, the Lipschitz condition*

$$\|\nabla h_\gamma(x) - \nabla h_\gamma(y)\| \;\le\; L_\gamma\,\|x - y\|, \qquad \forall\, x, y \in \mathbb{R}^d,$$

*where $L_\gamma > 0$ a.s. and $\mathbb{E}[L_\gamma^m] < \infty$ for every $m \ge 1$.*

*Then $\{X_t : t \in [0,T]\}$ is an order-$(1,1)$ weak approximation of $\{x_k\}$, namely: for each $g \in G^2$, there exists a constant $C > 0$, independent of $\eta, \rho$, such that*

$$\max_{0 \le k \le N}\Big|\,\mathbb{E}\big[g(x_k)\big] - \mathbb{E}\big[g(X_{k\eta})\big]\Big| \;\le\; C\,\big(\eta + \rho^2\big).$$

*Proof Sketch.* Theorem B.2 follows by replacing the single-parameter Lemmas A.1, A.2 and A.5 in Compagnoni et al. (2023) with our two-parameter versions—Theorem A.2, Lemma A.4 and Lemma A.5—imposing the extra global Lipschitz conditions to guarantee existence and uniqueness of the strong solution, and using our Assumption 3.1 to expand the drift term. We therefore omit the routine algebraic details. □

## B.2 n-USAM

For the n-USAM algorithm, we define the continuous-time approximation $X_t$ as the solution to the following SDE:

$$dX_t = -\nabla f^{n\text{-}USAM}(X_t)\,dt + \sqrt{\eta\,\Sigma^{n\text{-}USAM}(X_t)}\,dW_t, \tag{30}$$

where

$$f^{n\text{-}USAM}(X_t) := f(X_t) + \frac{\rho}{2}\|\nabla f(X_t)\|^2$$

$$\Sigma^{n\text{-}USAM}(X_t) := \Sigma_{0,0}(X_t) + \rho\big(\Sigma_{0,1}(X_t) + \Sigma_{0,1}^\top(X_t)\big). \tag{31}$$

$$\Sigma_{0,0}(X_t) := \mathbb{E}\left[\big(\nabla f_\gamma(X_t) - \nabla f(X_t)\big)\big(\nabla f_\gamma(X_t) - \nabla f(X_t)\big)^\top\right]$$

$$\Sigma_{0,1}(X_t) := \mathbb{E}\big[\big(\nabla f_\gamma(X_t) - \nabla f(X_t)\big) \cdot \big(\big(\nabla^2 f_\gamma(X_t) - \nabla^2 f(X_t)\big)\nabla f(X_t)\big)^\top\big].$$

We begin by deriving, via the following lemma, a one-step error estimate for the n-USAM discrete algorithm, which will be used to prove the main approximation theorem.

**Lemma B.3** (One-step moment estimates for n-USAM up to $\eta^1, \rho^1$). *Under Assumptions 3.1 and B.1. Define*

$$\partial_i f^{\text{n-USAM}}(x) := \partial_i f(x) \;+\; \rho \sum_j \partial_{ij}^2 f(x)\,\partial_j f(x),$$

*Let $\Delta(x)$ be the one-step increment defined in (24). Then:*

*(i) $\mathbb{E}\big[\Delta_{(i)}(x)\big] = -\,\partial_i f^{\text{n-USAM}}(x)\,\eta + O(\eta\,\rho^2).$*

*(ii)* $\mathbb{E}\big[\Delta_{(i)}(x)\,\Delta_{(j)}(x)\big] \quad = \quad \eta^2\Big(\partial_i f(x)\,\partial_j f(x) \quad + \quad \Sigma^{\text{n-USAM}}_{(ij)}(x)\Big) \quad +$

$$\eta^2 \rho\Big(\partial_i f(x) \sum_{l=1}^{d} \partial^2_{jl} f(x)\,\partial_l f(x) + \partial_j f(x) \sum_{l=1}^{d} \partial^2_{il} f(x)\,\partial_l f(x)\Big) + O(\eta^2 \rho^2).$$

*(iii)* $\mathbb{E}\Big[\prod_{j=1}^{3} |\Delta_{(i_j)}(x)|\Big] = O(\eta^3).$

*Proof.* Recall that the n-USAM update is

$$x_{k+1} = x_k \;-\; \eta\,\nabla f_{\gamma_k}\big(x_k + \rho\,\nabla f(x_k)\big),$$

so the one-step increment

$$\Delta(x) \;=\; x_{k+1} - x_k = \eta h(x,\gamma,\rho),$$

where we define

$$h(x,\gamma,\rho) := -\nabla f_\gamma\big(x + \rho\,\nabla f(x)\big).$$

By Taylor's theorem with integral remainder (Folland, 2005) we have, for each $\gamma$,

$$\nabla f_\gamma\big(x + \rho\,\nabla f(x)\big) = \nabla f_\gamma(x) + \rho\,\nabla^2 f_\gamma(x)\,\nabla f(x) + R(x,\gamma,\rho),$$

where

$$R(x,\gamma,\rho) = \int_0^1 (1-t)\,D^3 f_\gamma\big(x + t\,\rho\,\nabla f(x)\big)\big[\rho\,\nabla f(x),\,\rho\,\nabla f(x)\big]\,dt.$$

Here $D^3 f_\gamma(y)$ denotes the third-order tensor of partial derivatives of $f_\gamma$ at $y$, and $D^3 f_\gamma(y)[u,v]$ its bilinear action on vectors $u, v$.

Because $f_\gamma \in G^3$, there exists a polynomially bounded function $K(x) \in G$ such that

$$\big\|D^3 f_\gamma(y)\big\| \;\leq\; K(x), \quad \forall y \text{ with } \|y - x\| \leq \rho\,\|\nabla f(x)\|.$$

Hence

$$\big\|R(x,\gamma,\rho)\big\| \;\leq\; \int_0^1 (1-t)\,\big\|D^3 f_\gamma(x+t\rho\nabla f(x))\big\|\,\big\|\rho\nabla f(x)\big\|^2\,dt \;\leq\; K(x)\,\frac{\rho^2}{2}\,\|\nabla f(x)\|^2 = O\big(\rho^2\big),$$

uniformly in $\gamma$. Accordingly, a Taylor expansion in $\rho$ gives

$$\nabla f_\gamma(x + \rho\nabla f(x)) = \nabla f_\gamma(x) + \rho\,\nabla^2 f_\gamma(x)\,\nabla f(x) + O(\rho^2).$$

Hence

$$h(x,\gamma,\rho) = h_0(x,\gamma) \;+\; \rho\,h_1(x,\gamma) \;+\; O(\rho^2),$$

with

$$h_0(x,\gamma) := -\nabla f_\gamma(x), \qquad h_1(x,\gamma) := -\nabla^2 f_\gamma(x)\,\nabla f(x).$$

By Assumption 3.1 and Assumption B.1, each $h_0, h_1 \in G^2$, and

$$\mathbb{E}[h_0(x,\gamma)] = -\nabla f(x), \quad \mathbb{E}[h_1(x,\gamma)] = -\nabla^2 f(x)\,\nabla f(x).$$

We may therefore apply Lemma A.5 with these $h_0, h_1$, which yields exactly the three moment expansions up to $\eta^1, \rho^1$. $\qquad\square$

In Lemma B.3, we derived one-step moment estimates for the n-USAM discrete algorithm and, via Lemma A.4, for its corresponding SDE update (26). These estimates demonstrate that the first- and second-order moments satisfy the matching conditions of Theorem A.2. Together with the uniform moment bounds from Lemma D.2, we are now ready to establish the main weak-approximation theorem for n-USAM.

**Theorem B.4** (n-USAM SDE). *Under Assumptions 3.1 and B.1, let $0 < \eta < 1$, $T > 0$, and $N = \lfloor T/\eta \rfloor$. Denote by $\{x_k\}_{k=0}^{N}$ the n-USAM iterates in (7), and let $\{X_t\}_{t\in[0,T]}$ be the solution of the SDE (30). Suppose:*

*(i) The functions*

$$\nabla f^{\text{n-USAM}} \;=\; \nabla\Big(f + \tfrac{\rho}{2}\|\nabla f\|^2\Big) \quad and \quad \sqrt{\Sigma^{\text{n-USAM}}}$$

*are Lipschitz on $\mathbb{R}^d$.*

*(ii) The mapping*

$$h_\gamma(x) \;=\; -\nabla f_\gamma\big(x + \rho\,\nabla f(x)\big)$$

*satisfies, almost surely, the Lipschitz condition*

$$\|\nabla h_\gamma(x) - \nabla h_\gamma(y)\| \;\le\; L_\gamma\,\|x - y\|, \qquad \forall\, x, y \in \mathbb{R}^d,$$

*where $L_\gamma > 0$ a.s. and $\mathbb{E}[L_\gamma^m] < \infty$ for every $m \ge 1$.*

*Then $\{X_t : t \in [0, T]\}$ is an order-$(1,1)$ weak approximation of $\{x_k\}$, namely: for each $g \in G^2$, there exists a constant $C > 0$, independent of $\eta, \rho$, such that*

$$\max_{0 \le k \le N} \Big| \mathbb{E}\big[g(x_k)\big] - \mathbb{E}\big[g(X_{k\eta})\big] \Big| \;\le\; C\left(\eta + \rho^2\right).$$

*Proof.* First, we verify that SDE (30) admits a unique strong solution. By assumption, both the drift and diffusion coefficients are globally Lipschitz, which in turn implies a linear-growth condition. Therefore, Theorem D.1 applies and yields the existence and uniqueness of a strong solution on $[0, T]$.

Then, by Lemmas A.4, B.3, and D.2, all the conditions of Theorem A.2 are satisfied, and the proof is complete. $\qquad\square$

*Remark* B.5. The Lipschitz conditions are to ensure that the SDE has a unique strong solution with uniformly bounded moments. It is possible to appropriately relax them if we allow weak solutions (Mil'shtein, 1986).

## B.3 m-USAM

For the m-USAM algorithm, we define the continuous-time approximation $X_t$ as the solution to the following SDE:

$$dX_t = -\nabla\Big(f(X_t) + \tfrac{\rho}{2}\|\nabla f(X_t)\|^2 + \tfrac{\rho}{2m}\mathrm{tr}(V(X_t))\Big)dt + \sqrt{\tfrac{m\eta}{|\gamma|}}\Sigma^{m-USAM}(X_t)dW_t, \quad (32)$$

where

$$\Sigma^{m-USAM}(X_t) := \Sigma_{0,0}(X_t) + \rho(\Sigma_{0,1}(X_t) + \Sigma_{0,1}(X_t)^\top), \tag{33}$$

$$\Sigma_{0,0}(X_t) := \mathbb{E}\left[\big(\nabla f_{\mathcal{I}}(X_t) - \nabla f(X_t)\big)\big(\nabla f_{\mathcal{I}}(X_t) - \nabla f(X_t)\big)^\top\right],$$

$$\Sigma_{0,1}(X_t) := \mathbb{E}\big[\big(\nabla f_{\mathcal{I}}(X_t) - \nabla f(X_t)\big) \cdot \big(\nabla^2 f_{\mathcal{I}}(X_t)\nabla f_{\mathcal{I}}(X_t) - \mathbb{E}[\nabla^2 f_{\mathcal{I}}(X_t)\nabla f_{\mathcal{I}}(X_t)]\big)^\top\big].$$

We begin by deriving, via the following lemma, a one-step error estimate for the m-USAM discrete algorithm, which will be used to prove the main approximation theorem.

**Lemma B.6** (One-step moment estimates for m-USAM up to $\eta^1, \rho^1$)**.** *Under Assumptions 3.1 and B.1. Define*

$$\partial_i f^{m-USAM}(x) := \partial_i f(x) + \rho\mathbb{E}\left[\sum_{j=1}^{d} \partial_{ij}^2 f_{\mathcal{I}}(x)\partial_j f_{\mathcal{I}}(x)\right].$$

*Let $\Delta(x)$ be the one-step increment defined in* (24). *Then:*

*(i)* $\mathbb{E}\big[\Delta_{(i)}(x)\big] = -\partial_i f^{\text{m-USAM}}(x)\,\eta + O(\eta\rho^2).$

*(ii)* $\mathbb{E}\big[\Delta_{(i)}(x)\Delta_{(j)}(x)\big] \qquad = \qquad \eta^2\Big(\partial_i f(x)\,\partial_j f(x) \qquad + \qquad \Sigma_{(ij)}^{\text{m-USAM}}(x)\Big) \qquad +$

$\eta^2\rho\Big(\partial_i f(x)\sum_{l=1}^{d} \partial_{jl}^2 f_{\mathcal{I}}(x)\,\partial_l f_{\mathcal{I}}(x) + \partial_j f(x)\sum_{l=1}^{d} \partial_{il}^2 f_{\mathcal{I}}(x)\,\partial_l f_{\mathcal{I}}(x)\Big) + O(\eta^2\rho^2).$

*(iii)* $\mathbb{E}\left[\prod_{j=1}^{3} |\Delta_{(i_j)}(x)|\right] = O(\eta^3).$

*Proof.* Recall that the m-USAM update is

$$x_{k+1} = x_k - \frac{\eta\,m}{|\gamma|} \sum_{\mathcal{I}_j \subset \gamma_k, |\mathcal{I}_j|=m} \nabla f_{\mathcal{I}_j}\big(x_k + \rho\,\nabla f_{\mathcal{I}_j}(x_k)\big)$$

so the one-step increment

$$\Delta(x) \;=\; x_{k+1} - x_k = \eta h(x, \gamma, \rho),$$

where we define

$$h(x, \gamma, \rho) := -\frac{m}{|\gamma|} \sum_{\mathcal{I}_j \subset \gamma_k, |\mathcal{I}_j|=m} \nabla f_{\mathcal{I}_j}\big(x_k + \rho\,\nabla f_{\mathcal{I}_j}(x_k)\big).$$

By Taylor's theorem with integral remainder (Folland, 2005) we have, for each $\gamma$,

$$\sum_{\substack{\mathcal{I}_j \subset \gamma_k, \\ |\mathcal{I}_j|=m}} \nabla f_{\mathcal{I}_j}\big(x_k + \rho\,\nabla f_{\mathcal{I}_j}(x_k)\big) = \sum_{\substack{\mathcal{I}_j \subset \gamma_k, \\ |\mathcal{I}_j|=m}} \nabla f_{\mathcal{I}_j}(x_k) + \rho\,\nabla^2 f_{\mathcal{I}_j}(x_k)\nabla f_{\mathcal{I}_j}(x_k) + R(x, \gamma, \rho)$$

where

$$R(x, \gamma, \rho) = \int_0^1 (1-t)\, D^3 f_{\mathcal{I}_j}\big(x + t\,\rho\,\nabla f_{\mathcal{I}_j}(x)\big)\big[\rho\,\nabla f_{\mathcal{I}_j}(x),\, \rho\,\nabla f_{\mathcal{I}_j}(x)\big]\, dt.$$

Here $D^3 f_{\mathcal{I}_j}(y)$ denotes the third-order tensor of partial derivatives of $f_{\mathcal{I}_j}$ at $y$, and $D^3 f_{\mathcal{I}_j}(y)[u, v]$ its bilinear action on vectors $u, v$.

Because $f_{\mathcal{I}_j} \in G^3$, there exists a polynomially bounded function $K(x) \in G$ such that

$$\big\| D^3 f_{\mathcal{I}_j}(y) \big\| \;\leq\; K(x), \quad \forall y \text{ with } \|y - x\| \leq \rho \,\|\nabla f_{\mathcal{I}_j}(x)\|.$$

Hence

$$\big\| R(x, \gamma, \rho) \big\| \;\leq\; \int_0^1 (1-t)\, \big\| D^3 f_{\mathcal{I}_j}(x + t\rho\nabla f_{\mathcal{I}_j}(x)) \big\| \, \big\| \rho\nabla f_{\mathcal{I}_j}(x) \big\|^2 \, dt \;\leq\; K(x)\, \frac{\rho^2}{2}\, \|\nabla f_{\mathcal{I}_j}(x)\|^2$$

$$= O\big(\rho^2\big),$$

uniformly in $\gamma$. Accordingly, a Taylor expansion in $\rho$ gives

$$\nabla f_\gamma(x + \rho\nabla f(x)) = \nabla f_\gamma(x) + \rho\,\nabla^2 f_\gamma(x)\,\nabla f(x) + O(\rho^2).$$

Hence

$$h(x, \gamma, \rho) = h_0(x, \gamma) \;+\; \rho\, h_1(x, \gamma) \;+\; O(\rho^2),$$

with

$$h_0(x, \gamma) := -\frac{m}{|\gamma|} \sum_{\substack{\mathcal{I}_j \subset \gamma_k, \\ |\mathcal{I}_j|=m}} \nabla f_{\mathcal{I}_j}(x_k), \qquad h_1(x, \gamma) := -\frac{m}{|\gamma|} \sum_{\substack{\mathcal{I}_j \subset \gamma_k, \\ |\mathcal{I}_j|=m}} \nabla^2 f_{\mathcal{I}_j}(x_k)\nabla f_{\mathcal{I}_j}(x_k).$$

By Assumption 3.1 and Assumption B.1, each $h_0, h_1 \in G^2$, and

$$\mathbb{E}[h_0(x, \gamma)] = -\nabla f(x), \quad \mathbb{E}[h_1(x, \gamma)] = -\mathbb{E}\big[\nabla^2 f_{\mathcal{I}_j}(x)\,\nabla f_{\mathcal{I}_j}(x)\big].$$

We may therefore apply Lemma A.5 with these $h_0, h_1$, which yields exactly the three moment expansions up to $\eta^1, \rho^1$. $\qquad\square$

In Lemma B.6, we derived one-step moment estimates for the m-USAM discrete algorithm and, via Lemma A.4, for its corresponding SDE update (27). These estimates demonstrate that the first- and second-order moments satisfy the matching conditions of Theorem A.2. Together with the uniform moment bounds from Lemma D.2, we are now ready to establish the main weak-approximation theorem for m-USAM.

**Theorem B.7** (m-USAM SDE). *Under Assumptions 3.1 and B.1, let $0 < \eta < 1$, $T > 0$, and $N = \lfloor T/\eta \rfloor$. Denote by $\{x_k\}_{k=0}^N$ the m-USAM iterates in (8), and let $\{X_t\}_{t \in [0,T]}$ be the solution of the SDE (32). Suppose:*

*(i) The functions*

$$\nabla f^{\text{m-USAM}} \;=\; \nabla\Big(f + \frac{\rho}{2}\|\nabla f\|^2 + \frac{\rho}{2m}\text{tr}(V)\Big) \quad \text{and} \quad \sqrt{\Sigma^{\text{m-USAM}}}$$

*are Lipschitz on $\mathbb{R}^d$.*

*(ii) The mapping*

$$h_\gamma(x) \;=\; -\frac{m}{|\gamma|} \sum_{\mathcal{I}_j \subset \gamma, |\mathcal{I}_j|=m} \nabla f_{\mathcal{I}_j}\big(x + \rho\,\nabla f_{\mathcal{I}_j}(x)\big)$$

*satisfies, almost surely, the Lipschitz condition*

$$\|\nabla h_\gamma(x) - \nabla h_\gamma(y)\| \;\leq\; L_\gamma \, \|x - y\|, \qquad \forall\, x, y \in \mathbb{R}^d,$$

*where $L_\gamma > 0$ a.s. and $\mathbb{E}[L_\gamma^m] < \infty$ for every $m \geq 1$.*

*Then $\{X_t : t \in [0,T]\}$ is an order-$(1,1)$ weak approximation of $\{x_k\}$, namely: for each $g \in G^2$, there exists a constant $C > 0$, independent of $\eta, \rho$, such that*

$$\max_{0 \leq k \leq N} \Big| \mathbb{E}\big[g(x_k)\big] - \mathbb{E}\big[g(X_{k\eta})\big] \Big| \;\leq\; C\,(\eta + \rho^2).$$

*Proof.* First, we verify that SDE (32) admits a unique strong solution. By assumption, both the drift and diffusion coefficients are globally Lipschitz, which in turn implies a linear-growth condition. Therefore, Theorem D.1 applies and yields the existence and uniqueness of a strong solution on $[0, T]$.

Then, by Lemmas A.4, B.6, and D.2, all the conditions of Theorem A.2 are satisfied, and the proof is complete. $\qquad \square$

*Remark* B.8. The Lipschitz conditions are to ensure that the SDE has a unique strong solution with uniformly bounded moments. It is possible to appropriately relax them if we allow weak solutions (Mil'shtein, 1986).

## C   SDE approximation for SAM variants

Recall that the update rules of SAM variants are defined by:

$$\text{mini-batch SAM:} \quad x_{k+1} = x_k - \eta\,\nabla f_{\gamma_k}\Big(x_k + \rho\,\frac{\nabla f_{\gamma_k}(x_k)}{\|\nabla f_{\gamma_k}(x_k)\|}\Big) \tag{34}$$

$$\text{n-SAM:} \quad x_{k+1} = x_k - \eta\,\nabla f_{\gamma_k}\Big(x_k + \rho\,\frac{\nabla f(x_k)}{\|\nabla f(x_k)\|}\Big) \tag{35}$$

$$\text{m-SAM:} \quad x_{k+1} = x_k - \frac{\eta\,m}{|\gamma|} \sum_{\mathcal{I}_j \subset \gamma_k, |\mathcal{I}_j|=m} \nabla f_{\mathcal{I}_j}\Big(x_k + \rho\,\frac{\nabla f_{\mathcal{I}_j}(x_k)}{\|\nabla f_{\mathcal{I}_j}(x_k)\|}\Big) \tag{36}$$

### C.1   Mini-batch SAM

For the mini-batch SAM algorithm (34), we define the continuous-time approximation $X_t$ as the solution to the following SDE:

$$dX_t = -\nabla f^{SAM}(X_t)\,dt + \sqrt{\eta\,\Sigma^{SAM}(X_t)}\,dW_t, \tag{37}$$

where

$$f^{SAM}(X_t) := f(X_t) + \rho\,\mathbb{E}\|\nabla f_\gamma(X_t)\|$$
$$\Sigma^{SAM}(X_t) := \Sigma_{0,0}(X_t) + \rho\big(\Sigma_{0,1}(X_t) + \Sigma_{0,1}^\top(X_t)\big). \tag{38}$$

$$\Sigma_{0,0}(X_t) := \mathbb{E}\left[\left(\nabla f_\gamma(X_t) - \nabla f(X_t)\right)\left(\nabla f_\gamma(X_t) - \nabla f(X_t)\right)^\top\right]$$

$$\Sigma_{0,1}(X_t) := \mathbb{E}\left[\left(\nabla f_\gamma(X_t) - \nabla f(X_t)\right) \cdot \left(\frac{\nabla^2 f_\gamma(X_t)\nabla f_\gamma(X_t)}{\|\nabla f_\gamma(X_t)\|} - \mathbb{E}\left[\frac{\nabla^2 f_\gamma(X_t)\nabla f_\gamma(X_t)}{\|\nabla f_\gamma(X_t)\|}\right]\right)^\top\right].$$

*Remark* C.1 (On normalization at critical points). Note that SAM is ill-defined when the gradient is zero. To solve this, we may replace the denominator by $\|\cdot\|_\varepsilon = \sqrt{\|\cdot\|^2 + \varepsilon^2}$ with a fixed $\varepsilon > 0$, which is also a common implementation in practice. Under the standing assumption that $\nabla f$ is $L$-Lipschitz, the resulting coefficients are globally $O(L/\varepsilon)$-Lipschitz and $C^1$. All local Taylor or weak-approximation arguments and moment bounds in Appendix C continue to hold with constants depending on $\varepsilon$ but independent of $\eta, \rho$. The stated orders in $\eta, \rho$ are unaffected.

**Theorem C.2** (mini-batch SAM SDE, adapted from Theorem 3.5 of Compagnoni et al. (2023)).
*Under Assumptions 3.1 and B.1, let $0 < \eta < 1$, $T > 0$, and $N = \lfloor T/\eta \rfloor$. Denote by $\{x_k\}_{k=0}^N$ the mini-batch SAM iterates in (4), and let $\{X_t\}_{t\in[0,T]}$ be the solution of the SDE (37). Suppose:*

*(i) The functions*

$$\nabla f^{\mathrm{SAM}} = \nabla\left(f + \rho\mathbb{E}\|\nabla f_\gamma\|\right) \quad \text{and} \quad \sqrt{\Sigma^{\mathrm{SAM}}}$$

*are Lipschitz on $\mathbb{R}^d$.*

*(ii) The mapping*

$$h_\gamma(x) = -\nabla f_\gamma\left(x + \rho\frac{\nabla f_\gamma(x)}{\|\nabla f_\gamma(x)\|}\right)$$

*satisfies, almost surely, the Lipschitz condition*

$$\|\nabla h_\gamma(x) - \nabla h_\gamma(y)\| \leq L_\gamma \|x - y\|, \qquad \forall\, x, y \in \mathbb{R}^d,$$

*where $L_\gamma > 0$ a.s. and $\mathbb{E}[L_\gamma^m] < \infty$ for every $m \geq 1$.*

*Then $\{X_t : t \in [0, T]\}$ is an order-$(1, 1)$ weak approximation of $\{x_k\}$, namely: for each $g \in G^2$, there exists a constant $C > 0$, independent of $\eta$, such that*

$$\max_{0 \leq k \leq N}\left|\mathbb{E}\left[g(x_k)\right] - \mathbb{E}\left[g(X_{k\eta})\right]\right| \leq C\left(\eta + \rho^2\right).$$

*Proof Sketch.* Theorem C.2 follows by replacing the single-parameter Lemmas A.1, A.2 and A.14 in Compagnoni et al. (2023) with our two-parameter versions—Theorem A.2, Lemma A.4 and Lemma A.5—imposing the extra global Lipschitz conditions to guarantee existence and uniqueness of the strong solution. We therefore omit the routine algebraic details. $\qquad\square$

### C.2 n-SAM

For the n-SAM algorithm, we define the continuous-time approximation $X_t$ as the solution to the following SDE:

$$dX_t = -\nabla f^{n\text{-}SAM}(X_t)\,dt + \sqrt{\eta\,\Sigma^{n\text{-}SAM}(X_t)}\,dW_t, \tag{39}$$

where

$$f^{n\text{-}SAM}(X_t) := f(X_t) + \rho\|\nabla f(X_t)\|$$

$$\Sigma^{n\text{-}SAM}(X_t) := \Sigma_{0,0}(X_t) + \rho\left(\Sigma_{0,1}(X_t) + \Sigma_{0,1}^\top(X_t)\right). \tag{40}$$

$$\Sigma_{0,0}(X_t) := \mathbb{E}\left[\left(\nabla f_\gamma(X_t) - \nabla f(X_t)\right)\left(\nabla f_\gamma(X_t) - \nabla f(X_t)\right)^\top\right]$$

$$\Sigma_{0,1}(X_t) := \mathbb{E}\left[\left(\nabla f_\gamma(X_t) - \nabla f(X_t)\right) \cdot \left(\left(\nabla^2 f_\gamma(X_t) - \nabla^2 f(X_t)\right)\frac{\nabla f(X_t)}{\|\nabla f(X_t)\|}\right)^\top\right].$$

We begin by deriving, via the following lemma, a one-step error estimate for the n-SAM discrete algorithm, which will be used to prove the main approximation theorem.

**Lemma C.3** (One-step moment estimates for n-SAM up to $\eta^1, \rho^1$). *Under Assumptions 3.1 and B.1, define*

$$\partial_i f^{\text{n-SAM}}(x) := \partial_i f(x) \ + \ \rho \sum_{j=1}^{d} \partial_{ij}^2 f(x) \, \frac{\partial_j f(x)}{\|\nabla f(x)\|}.$$

*Let $\Delta(x)$ be the one-step increment defined in* (24). *Then:*

*(i)* $\mathbb{E}\big[\Delta_{(i)}(x)\big] = - \, \partial_i f^{\text{n-SAM}}(x) \, \eta + O(\eta \, \rho^2).$

*(ii)* $\mathbb{E}\big[\Delta_{(i)}(x) \, \Delta_{(j)}(x)\big] \qquad = \qquad \eta^2 \Big(\partial_i f(x) \, \partial_j f(x) \quad + \quad \Sigma_{(ij)}^{\text{n-SAM}}(x)\Big) \qquad +$

$\eta^2 \, \rho \, \Big(\partial_i f(x) \sum_{l=1}^{d} \partial_{jl}^2 f(x) \, \frac{\partial_l f(x)}{\|\nabla f(x)\|} + \partial_j f(x) \sum_{l=1}^{d} \partial_{il}^2 f(x) \, \frac{\partial_l f(x)}{\|\nabla f(x)\|}\Big) + O(\eta^2 \, \rho^2).$

*(iii)* $\mathbb{E}\Big[\prod_{j=1}^{3} |\Delta_{(i_j)}(x)|\Big] = O(\eta^3).$

*Proof.* The n-SAM update is

$$x_{k+1} = x_k \ - \ \eta \, \nabla f_{\gamma_k}\Big(x_k + \rho \, \tfrac{\nabla f(x_k)}{\|\nabla f(x_k)\|}\Big),$$

so

$$\Delta(x) = x_{k+1} - x_k = \eta \, h\big(x, \gamma, \rho\big),$$

with

$$h(x, \gamma, \rho) := - \nabla f_\gamma\big(x + \rho \, u(x)\big), \qquad u(x) := \frac{\nabla f(x)}{\|\nabla f(x)\|}.$$

By Taylor's theorem with integral remainder (Folland, 2005), for each $\gamma$ and writing $u(x) = \nabla f(x)/\|\nabla f(x)\|$, we have

$$\nabla f_\gamma\big(x + \rho \, u(x)\big) = \nabla f_\gamma(x) + \rho \, \nabla^2 f_\gamma(x) \, u(x) + R(x, \gamma, \rho),$$

where

$$R(x, \gamma, \rho) = \int_0^1 (1 - t) \, D^3 f_\gamma\big(x + t \, \rho \, u(x)\big)\big[\rho \, u(x), \, \rho \, u(x)\big] \, dt.$$

Here $D^3 f_\gamma(y)$ denotes the third-order tensor of partial derivatives of $f_\gamma$ at $y$, and $D^3 f_\gamma(y)[v, w]$ its bilinear action on vectors $v, w$.

Because $f_\gamma \in G^3$, there exists a polynomially bounded function $K(x) \in G$ such that

$$\big\|D^3 f_\gamma(y)\big\| \ \leq \ K(x), \quad \forall y \text{ with } \|y - x\| \leq \rho \, \|u(x)\|.$$

Hence

$$\|R(x, \gamma, \rho)\| \leq \int_0^1 (1 - t) \, K(x) \, \|\rho \, u(x)\|^2 \, dt = O(\rho^2),$$

uniformly in $\gamma$. Hence

$$h(x, \gamma, \rho) = h_0(x, \gamma) + \rho \, h_1(x, \gamma) + O(\rho^2),$$

with

$$h_0(x, \gamma) := -\nabla f_\gamma(x), \qquad h_1(x, \gamma) := - \nabla^2 f_\gamma(x) \, \frac{\nabla f(x)}{\|\nabla f(x)\|}.$$

By Assumptions 3.1 and B.1, each $h_0, h_1 \in G^2$ and

$$\mathbb{E}[h_0(x, \gamma)] = -\nabla f(x), \qquad \mathbb{E}[h_1(x, \gamma)] = - \nabla^2 f(x) \, \frac{\nabla f(x)}{\|\nabla f(x)\|}.$$

The same application of Lemma A.5 then yields the moment expansions (i)–(iii) up to order $\eta^1, \rho^1$ as stated. $\qquad\square$

In Lemma C.3, we derived one-step moment estimates for the n-SAM discrete algorithm and, via Lemma A.4, for its corresponding SDE update (35). These estimates demonstrate that the first- and second-order moments satisfy the matching conditions of Theorem A.2. Together with the uniform moment bounds from Lemma D.2, we are now ready to establish the main weak-approximation theorem for n-SAM.

**Theorem C.4** (n-SAM SDE). *Under Assumptions 3.1 and B.1, let $0 < \eta < 1$, $T > 0$, and $N = \lfloor T/\eta \rfloor$. Denote by $\{x_k\}_{k=0}^N$ the n-SAM iterates in (3), and let $\{X_t\}_{t \in [0,T]}$ be the solution of the SDE (39). Suppose:*

*(i) The functions*

$$\nabla f^{\text{n-SAM}} = \nabla\Big(f + \rho\|\nabla f\|\Big) \quad and \quad \sqrt{\Sigma^{\text{n-SAM}}}$$

*are Lipschitz on $\mathbb{R}^d$.*

*(ii) The mapping*

$$h_\gamma(x) = -\nabla f_\gamma\big(x + \rho \frac{\nabla f(x)}{\|\nabla f(x)\|}\big)$$

*satisfies, almost surely, the Lipschitz condition*

$$\|\nabla h_\gamma(x) - \nabla h_\gamma(y)\| \leq L_\gamma \|x - y\|, \qquad \forall\, x, y \in \mathbb{R}^d,$$

*where $L_\gamma > 0$ a.s. and $\mathbb{E}[L_\gamma^m] < \infty$ for every $m \geq 1$.*

*Then $\{X_t : t \in [0, T]\}$ is an order-$(1,1)$ weak approximation of $\{x_k\}$, namely: for each $g \in G^2$, there exists a constant $C > 0$, independent of $\eta$, such that*

$$\max_{0 \leq k \leq N} \Big| \mathbb{E}\big[g(x_k)\big] - \mathbb{E}\big[g(X_{k\eta})\big] \Big| \leq C\,(\eta + \rho^2).$$

*Proof.* First, we verify that SDE (39) admits a unique strong solution. By assumption, both the drift and diffusion coefficients are globally Lipschitz, which in turn implies a linear-growth condition. Therefore, Theorem D.1 applies and yields the existence and uniqueness of a strong solution on $[0, T]$.

Then, by Lemmas A.4, C.3, and D.2, all the conditions of Theorem A.2 are satisfied, and the proof is complete. $\qquad\square$

*Remark* C.5. The Lipschitz conditions are to ensure that the SDE has a unique strong solution with uniformly bounded moments. It is possible to appropriately relax them if we allow weak solutions (Mil'shtein, 1986).

## C.3 m-SAM

For the m-SAM algorithm, we define the continuous-time approximation $X_t$ as the solution to the following SDE:

$$dX_t = -\nabla\big(f(X_t) + \frac{\rho}{m}\mathbb{E}\|\sum_{\substack{i \in \mathcal{I}, \\ |\mathcal{I}|=m}} \nabla f_i(X_t)\|\big)dt \quad + \sqrt{\frac{m\eta}{|\gamma|}}\big(\Sigma^{m-SAM}(X_t)\big)^{\frac{1}{2}} dW_t, \qquad (41)$$

where

$$\Sigma^{m-SAM}(X_t) := \Sigma_{0,0}(X_t) + \rho(\Sigma_{0,1}(X_t) + \Sigma_{0,1}(X_t)^\top), \qquad (42)$$

$$\Sigma_{0,0}(X_t) := \mathbb{E}\left[\big(\nabla f_\mathcal{I}(X_t) - \nabla f(X_t)\big)\big(\nabla f_\mathcal{I}(X_t) - \nabla f(X_t)\big)^\top\right],$$

$$\Sigma_{0,1}(X_t) := \mathbb{E}\big[(\nabla f_\mathcal{I}(X_t) - \nabla f(X_t)) \cdot \big(\nabla^2 f_\mathcal{I}(X_t)\frac{\nabla f_\mathcal{I}(X_t)}{\|\nabla f_\mathcal{I}(X_t)\|} - \mathbb{E}[\nabla^2 f_\mathcal{I}(X_t)\frac{\nabla f_\mathcal{I}(X_t)}{\|\nabla f_\mathcal{I}(X_t)\|}]\big)^\top\big].$$

We begin by deriving, via the following lemma, a one-step error estimate for the m-sam discrete algorithm, which will be used to prove the main approximation theorem.

**Lemma C.6** (One-step moment estimates for m-SAM up to $\eta^1, \rho^1$)**.** *Under Assumptions 3.1 and B.1, define*

$$\partial_i f^{m\text{-}SAM}(x) := \partial_i f(x) + \rho\, \mathbb{E}\Big[\sum_{j=1}^{d} \partial_{ij}^2 f_{\mathcal{I}}(x)\, \frac{\partial_j f_{\mathcal{I}}(x)}{\|\nabla f_{\mathcal{I}}(x)\|}\Big].$$

*Let $\Delta(x)$ be the one-step increment defined in* (24). *Then:*

*(i)* $\mathbb{E}\big[\Delta_{(i)}(x)\big] = -\,\partial_i f^{m\text{-}SAM}(x)\,\eta + O(\eta\,\rho^2).$

*(ii)* $\mathbb{E}\big[\Delta_{(i)}(x)\,\Delta_{(j)}(x)\big] \quad = \quad \eta^2\Big(\partial_i f(x)\,\partial_j f(x) \quad + \quad \Sigma_{(ij)}^{m\text{-}SAM}(x)\Big) \quad +$

$\eta^2\,\rho\,\Big(\partial_i f(x)\quad \mathbb{E}\Big[\sum_{l=1}^{d} \partial_{jl}^2 f_{\mathcal{I}}(x)\,\frac{\partial_l f_{\mathcal{I}}(x)}{\|\nabla f_{\mathcal{I}}(x)\|}\Big] \quad + \quad \partial_j f(x)\quad \mathbb{E}\Big[\sum_{l=1}^{d} \partial_{il}^2 f_{\mathcal{I}}(x)\,\frac{\partial_l f_{\mathcal{I}}(x)}{\|\nabla f_{\mathcal{I}}(x)\|}\Big]\Big) \quad +$
$O(\eta^2\,\rho^2).$

*(iii)* $\mathbb{E}\Big[\prod_{j=1}^{3}|\Delta_{(i_j)}(x)|\Big] = O(\eta^3).$

*Proof.* Recall that the m-SAM update is

$$x_{k+1} = x_k - \frac{\eta\, m}{|\gamma|}\sum_{\mathcal{I}_j \subset \gamma_k,\, |\mathcal{I}_j|=m} \nabla f_{\mathcal{I}_j}\Big(x_k + \rho\,\frac{\nabla f_{\mathcal{I}_j}(x_k)}{\|\nabla f_{\mathcal{I}_j}(x_k)\|}\Big),$$

so the one-step increment

$$\Delta(x) = x_{k+1} - x_k = \eta\, h(x, \gamma, \rho),$$

where

$$h(x, \gamma, \rho) := -\frac{m}{|\gamma|}\sum_{\mathcal{I}_j \subset \gamma_k,\, |\mathcal{I}_j|=m} \nabla f_{\mathcal{I}_j}\Big(x + \rho\,\frac{\nabla f_{\mathcal{I}_j}(x)}{\|\nabla f_{\mathcal{I}_j}(x)\|}\Big).$$

By Taylor's theorem with integral remainder (Folland, 2005), for each subset index $\mathcal{I}_j$,

$$\nabla f_{\mathcal{I}_j}\Big(x + \rho\,\frac{\nabla f_{\mathcal{I}_j}(x)}{\|\nabla f_{\mathcal{I}_j}(x)\|}\Big) = \nabla f_{\mathcal{I}_j}(x) + \rho\,\nabla^2 f_{\mathcal{I}_j}(x)\,\frac{\nabla f_{\mathcal{I}_j}(x)}{\|\nabla f_{\mathcal{I}_j}(x)\|} + R(x, \gamma, \rho),$$

where

$$R(x, \gamma, \rho) = \int_0^1 (1-t)\, D^3 f_{\mathcal{I}_j}\Big(x + t\,\rho\,\frac{\nabla f_{\mathcal{I}_j}(x)}{\|\nabla f_{\mathcal{I}_j}(x)\|}\Big)\Big[\rho\,\frac{\nabla f_{\mathcal{I}_j}(x)}{\|\nabla f_{\mathcal{I}_j}(x)\|},\, \rho\,\frac{\nabla f_{\mathcal{I}_j}(x)}{\|\nabla f_{\mathcal{I}_j}(x)\|}\Big]\, dt.$$

Here $D^3 f_{\mathcal{I}_j}(y)$ is the third-order derivative tensor of $f_{\mathcal{I}_j}$ at $y$, and $D^3 f_{\mathcal{I}_j}(y)[u, v]$ its action on $(u, v)$. Since $f_{\mathcal{I}_j} \in G^3$, there is $K(x) \in G$ polynomially bounded so that

$$\|D^3 f_{\mathcal{I}_j}(y)\| \le K(x) \quad \text{whenever} \quad \|y - x\| \le \rho\,\Big\|\frac{\nabla f_{\mathcal{I}_j}(x)}{\|\nabla f_{\mathcal{I}_j}(x)\|}\Big\|.$$

Hence

$$\|R(x, \gamma, \rho)\| \le \int_0^1 (1-t)\, K(x)\,\|\rho\,\frac{\nabla f_{\mathcal{I}_j}(x)}{\|\nabla f_{\mathcal{I}_j}(x)\|}\|^2\, dt = \tfrac{1}{2}K(x)\,\rho^2 = O(\rho^2),$$

uniformly in $\gamma$. Thus a Taylor expansion in $\rho$ gives

$$h(x, \gamma, \rho) = h_0(x, \gamma) + \rho\, h_1(x, \gamma) + O(\rho^2),$$

with

$$h_0(x, \gamma) := -\frac{m}{|\gamma|}\sum_{\mathcal{I}_j \subset \gamma_k,\, |\mathcal{I}_j|=m} \nabla f_{\mathcal{I}_j}(x), \quad h_1(x, \gamma) := -\frac{m}{|\gamma|}\sum_{\mathcal{I}_j \subset \gamma_k,\, |\mathcal{I}_j|=m} \nabla^2 f_{\mathcal{I}_j}(x)\,\frac{\nabla f_{\mathcal{I}_j}(x)}{\|\nabla f_{\mathcal{I}_j}(x)\|}.$$

By Assumptions 3.1 and B.1, each $h_0, h_1 \in G^2$, and

$$\mathbb{E}[h_0(x, \gamma)] = -\nabla f(x), \quad \mathbb{E}[h_1(x, \gamma)] = -\mathbb{E}\Big[\nabla^2 f_{\mathcal{I}_j}(x)\,\frac{\nabla f_{\mathcal{I}_j}(x)}{\|\nabla f_{\mathcal{I}_j}(x)\|}\Big].$$

Applying Lemma A.5 to these $h_0, h_1$ yields exactly the three moment estimates up to $\eta^1, \rho^1$ as claimed. □

In Lemma C.6, we derived one-step moment estimates for the m-SAM discrete algorithm and, via Lemma A.4, for its corresponding SDE update (36). These estimates demonstrate that the first- and second-order moments satisfy the matching conditions of Theorem A.2. Together with the uniform moment bounds from Lemma D.2, we are now ready to establish the main weak-approximation theorem for m-SAM.

**Theorem C.7** (m-SAM SDE). *Under Assumptions 3.1 and B.1, let $0 < \eta < 1$, $T > 0$, and $N = \lfloor T/\eta \rfloor$. Denote by $\{x_k\}_{k=0}^N$ the m-SAM iterates in (5), and let $\{X_t\}_{t \in [0,T]}$ be the solution of the SDE (41). Suppose:*

*(i) The functions*

$$\nabla f^{\text{m-SAM}} = \nabla\big(f + \frac{\rho}{m}\mathbb{E}\|\sum_{\substack{i \in \mathcal{I}, \\ |\mathcal{I}|=m}} \nabla f_i\|\big) \quad and \quad \sqrt{\Sigma^{\text{m-SAM}}}$$

*are Lipschitz on $\mathbb{R}^d$.*

*(ii) The mapping*

$$h_\gamma(x) = -\frac{m}{|\gamma|} \sum_{\mathcal{I}_j \subset \gamma, |\mathcal{I}_j|=m} \nabla f_{\mathcal{I}_j}\big(x + \rho \frac{\nabla f_{\mathcal{I}_j}(x)}{\|\nabla f_{\mathcal{I}_j}(x)\|}\big)$$

*satisfies, almost surely, the Lipschitz condition*

$$\|\nabla h_\gamma(x) - \nabla h_\gamma(y)\| \leq L_\gamma \|x - y\|, \qquad \forall\, x, y \in \mathbb{R}^d,$$

*where $L_\gamma > 0$ a.s. and $\mathbb{E}[L_\gamma^m] < \infty$ for every $m \geq 1$.*

*Then $\{X_t : t \in [0,T]\}$ is an order-$(1,1)$ weak approximation of $\{x_k\}$, namely: for each $g \in G^2$, there exists a constant $C > 0$, independent of $\eta$, such that*

$$\max_{0 \leq k \leq N} \big| \mathbb{E}\big[g(x_k)\big] - \mathbb{E}\big[g(X_{k\eta})\big] \big| \leq C\,\big(\eta + \rho^2\big).$$

*Proof.* First, we verify that SDE (41) admits a unique strong solution. By assumption, both the drift and diffusion coefficients are globally Lipschitz, which in turn implies a linear-growth condition. Therefore, Theorem D.1 applies and yields the existence and uniqueness of a strong solution on $[0, T]$.

Then, by Lemmas A.4, C.6, and D.2, all the conditions of Theorem A.2 are satisfied, and the proof is complete. □

*Remark* C.8. The Lipschitz conditions are to ensure that the SDE has a unique strong solution with uniformly bounded moments. It is possible to appropriately relax them if we allow weak solutions (Mil'shtein, 1986).

# D   Auxiliary Lemmas

**Theorem D.1** (Existence and Uniqueness of Strong Solutions (Evans, 2012)). *Let $b : \mathbb{R}^d \to \mathbb{R}^d$ and $\sigma : \mathbb{R}^d \to \mathbb{R}^{d \times m}$ be measurable functions satisfying:*

*(i) **Global Lipschitz.** There exists a constant $L > 0$ such that*

$$\|b(x) - b(y)\| + \|\sigma(x) - \sigma(y)\| \leq L\|x - y\|, \quad \forall\, x, y \in \mathbb{R}^d.$$

*(ii) **Linear growth.** There exists a constant $K > 0$ such that*

$$\|b(x)\|^2 + \|\sigma(x)\|^2 \leq K\big(1 + \|x\|^2\big), \quad \forall\, x \in \mathbb{R}^d.$$

*Let $X_0$ be an $\mathbb{R}^d$-valued random variable with $\mathbb{E}[\|X_0\|^2] < \infty$. Then the SDE*

$$dX_t = b(X_t)\,dt + \sigma(X_t)\,dW_t, \quad X_0 \text{ given},$$

*admits a unique strong solution $\{X_t\}_{t \geq 0}$ satisfying*

$$\mathbb{E}\Big[\sup_{0 \leq s \leq T} \|X_s\|^2\Big] < \infty, \quad \forall\, T > 0.$$

**Lemma D.2.** *(Li et al., 2019) Let $\{x_k : k \geq 0\}$ be the generalized iterations defined in* (22)*. Suppose*

$$\big|h(x, \gamma, \eta)\big| \leq L_\gamma\big(1 + \|x\|\big),$$

*where $L_\gamma > 0$ almost surely and*

$$\mathbb{E}\big[L_\gamma^m\big] < \infty \quad \text{for all } m \geq 1.$$

*Then for any fixed $T > 0$ and any $m \geq 1$, the moment $\mathbb{E}\big[\|x_k\|^m\big]$ exists and is uniformly bounded in both $\eta$ and $k = 0, 1, \ldots, N$, where $N = \lfloor T/\eta \rfloor$.*

# E    Proof of Proposition 3.9

We will use the following lemma on convex order to prove Proposition 3.9, whose proof can be found in classical textbooks on stochastic order, such as Müller and Stoyan (2002).

**Lemma E.1** (Convex-order Monotonicity)*. Let $X_1, \ldots, X_n$ be i.i.d. random vectors in $\mathbb{R}^d$ with a log-concave density. For each integer $1 \leq k \leq n$, define*

$$S_k = \frac{1}{k} \sum_{i=1}^{k} X_i.$$

*Then for any $1 \leq k < m \leq n$ and any convex function $\phi \colon \mathbb{R}^d \to \mathbb{R}$,*

$$\mathbb{E}\big[\phi(S_m)\big] \leq \mathbb{E}\big[\phi(S_k)\big].$$

*Proof of Proposition 3.9.* Lower bound: Applying Jensen's inequality,

$$\|\nabla f(x)\| = \big\|\mathbb{E}[\nabla f_\gamma(x)]\big\| \leq \mathbb{E}[\|\nabla f_\gamma(x)\|].$$

Upper bound: By Cauchy–Schwarz inequality,

$$\mathbb{E}[\|\nabla f_\gamma(x)\|] \leq \sqrt{\mathbb{E}[\|\nabla f_\gamma(x)\|^2]} = \sqrt{\|\nabla f(x)\|^2 + \frac{\mathrm{tr}(V(x))}{|\gamma|}}.$$

Combining both bounds completes the proof of the first statement.

For the second statement, we apply Lemma E.1 to the convex function $\|\cdot\|$, which concludes the proof. $\qquad\square$

# F    Additional Related Works

**Theoretical understanding of SAM.** Although SAM and its variants have achieved remarkable success in various practical applications (Foret et al., 2021; Kwon et al., 2021; Kaddour et al., 2022; Li et al., 2024b; Li and Giannakis, 2024), the theoretical understanding behind them remains limited. The pioneering work of Andriushchenko and Flammarion (2022) provided the first theoretical framework for understanding SAM, covering its convergence properties and implicit bias in simple network structures, while also systematically illustrating several empirical phenomena. Subsequently, Si and Yun (2024) extended the convergence analysis to various deterministic and stochastic settings. Compagnoni et al. (2023); Luo et al. (2025) conducted an in-depth analysis of the dynamics of SAM using the SDE framework previously developed by Li et al. (2017), leading to a deeper understanding of its implicit bias. On the other hand, Wen et al. (2022) investigated the implicit bias of SAM by analyzing its slow ordinary differential equation (ODE) behavior near the minimizer manifold, demonstrating how SAM drifts toward flatter minima. More recently, Zhou et al. (2024) studied the late-stage behavior of SAM using stability analysis, showing its advantage in escaping sharp minima.

# G    Derivation of the Finite Difference Estimator in Equation (21)

We start with the first-order Taylor expansion around $x$:

$$\frac{f_i\big(x + \delta z\big) - f_i(x)}{\delta} = \nabla f_i(x)^\top z + O(\delta),$$

where $z \in \{\pm 1\}^d$ is a Rademacher random vector, and $f_i$ is assumed twice differentiable so that the remainder is of order $O(\delta)$.

**Step 1: Square both sides.**

$$\left(\tfrac{f_i(x+\delta z)-f_i(x)}{\delta}\right)^2 \;=\; \left(\nabla f_i(x)^\top z\right)^2 \;+\; 2\big(\nabla f_i(x)^\top z\big)\,O(\delta) \;+\; O(\delta)^2.$$

Often we simply write this as

$$\left(\nabla f_i(x)^\top z + O(\delta)\right)^2 \;=\; \left(\nabla f_i(x)^\top z\right)^2 \;+\; O(\delta)\left(\nabla f_i(x)^\top z\right) \;+\; O(\delta^2).$$

**Step 2: Take expectation over $z$.** Because $z$ has independent $\{\pm 1\}$ components, we have

$$\mathbb{E}_z\left[\left(\nabla f_i(x)^\top z\right)^2\right] \;=\; \left\|\nabla f_i(x)\right\|^2 \quad \text{(standard Rademacher property).}$$

Hence,

$$\mathbb{E}_z\left[\left(\tfrac{f_i(x+\delta z)-f_i(x)}{\delta}\right)^2\right] \;=\; \mathbb{E}_z\left[\left(\nabla f_i(x)^\top z\right)^2\right] \;+\; O(\delta^2) \;=\; \|\nabla f_i(x)\|^2 + O(\delta^2).$$

Thus the mean-squared estimate of the finite difference quotient differs from $\|\nabla f_i(x)\|^2$ by an $O(\delta^2)$ bias term, implying that the estimator is approximately unbiased as $\delta \to 0$. We compare two $d$-dimensional random vectors $z \in \mathbb{R}^d$:

- **Rademacher:** each component $z_j$ is independently $\pm 1$ with probability $1/2$,
- **Standard Gaussian:** each component $z_j$ is i.i.d. $\mathcal{N}(0,1)$.

Both have $\mathbb{E}[z_j] = 0$ and $\mathbb{E}[z_j^2] = 1$, so $\mathbb{E}[(z^\top v)^2] = \|v\|^2$ for any $v \in \mathbb{R}^d$. We look at the *fourth moment* $\mathbb{E}[(z^\top v)^4]$, relevant to the variance in many finite-difference or gradient estimators.

**Rademacher case.** Since $z_j^2 \equiv 1$,

$$(z^\top v)^2 = \Big(\sum_{j=1}^d v_j z_j\Big)^2 = \sum_{j,k} v_j v_k\, z_j z_k.$$

Then

$$(z^\top v)^4 = \Big(\sum_{j,k} v_j v_k\, z_j z_k\Big)^2$$

and using $\mathbb{E}[z_j^4] = 1$, $\mathbb{E}[z_j^2 z_k^2] = 1$ (when $j \neq k$) with zero cross terms of odd product, one obtains a relatively small constant factor. Detailed calculation yields

$$\mathbb{E}\big[(z^\top v)^4\big] = \sum_{j=1}^d v_j^4 + 6 \sum_{j<k} v_j^2 v_k^2 \leq 3\left\|v\right\|^4 \quad \text{(for } d > 1\text{).}$$

**Gaussian case.** If $z \sim \mathcal{N}(0, I_d)$, then $\mathbb{E}[z_j^4] = 3$ and $\mathbb{E}[z_j^2 z_k^2] = 1$ for $j \neq k$. One can show

$$\mathbb{E}[(z^\top v)^4] = 3\|v\|^4.$$

Hence the constant factor in front of $\|v\|^4$ is exactly 3 for Gaussian.

**Conclusion.** For both Rademacher and Gaussian vectors, $\mathbb{E}[(z^\top v)^2] = \|v\|^2$. However, when analyzing higher-order moments (e.g. $(z^\top v)^4$) that affect the variance of many finite-difference or random-direction estimators, the Rademacher distribution can yield a smaller constant factor. This often leads to reduced variance and tighter theoretical bounds for the same sample size.

## H   Derivation of the Gibbs Distribution (20)

Consider the following maximization problem:

$$\max_{P \in \Delta} \quad \sum_{i \in \gamma} p_i \, \|\nabla f_i(x)\| \;+\; \frac{\mathbb{H}(P)}{\lambda},$$

where $\Delta$ is the probability simplex (i.e., $\sum_i p_i = 1$ and $p_i \geq 0$), $\mathbb{H}(P) = -\sum_i p_i \ln p_i$ is the entropy term, and $\lambda > 0$ is a given constant.

1. Construct the Lagrangian. We introduce the constraint $\sum_i p_i = 1$ with a Lagrange multiplier $\alpha$:

$$\mathcal{L}(P, \alpha) = \sum_i p_i \|\nabla f_i(x)\| + \frac{1}{\lambda}\Big(-\sum_i p_i \ln p_i\Big) + \alpha\Big(1 - \sum_i p_i\Big).$$

2. Differentiate w.r.t. $p_i$ and set to zero. Taking partial derivatives with respect to $p_i$ and setting them to zero,

$$\frac{\partial \mathcal{L}}{\partial p_i} = \|\nabla f_i(x)\| - \frac{1}{\lambda}(\ln p_i + 1) - \alpha = 0.$$

Therefore,

$$\|\nabla f_i(x)\| - \frac{\ln p_i + 1}{\lambda} - \alpha = 0 \implies p_i = \exp\big(\lambda \|\nabla f_i(x)\| + 1 - \lambda\alpha\big).$$

3. Enforce the normalization. The constraint $\sum_i p_i = 1$ fixes the value of $\alpha$; it amounts to one overall normalization factor in the denominator. Hence the solution takes the well-known Gibbs distribution form:

$$p_i^* = \frac{\exp\big(\lambda \|\nabla f_i(x)\|\big)}{\sum_j \exp\big(\lambda \|\nabla f_j(x)\|\big)}.$$

# I   Algorithm: Reweighted SAM

---
**Algorithm 1** Reweighted SAM

---
1: **while** not converged **do**
2:     Forward pass to obtain $f_{\gamma_k}(x_k)$
3:     **for** $q = 1, \ldots, Q$ **do**
4:         Estimate per-sample gradient norm using Eq. (21)
5:     **end for**
6:     Normalize estimated per-sample gradient norm
7:     Compute weight $p^*$ using Eq. (20)
8:     Compute perturbation $\epsilon_k$ using Eq. (18)
9:     Compute perturbed gradient $g_k = \nabla f_{\gamma_k}(x_k + \rho\epsilon_k)$
10:     Update model parameters: $x_{k+1} = x_k - \eta g_k$
11: **end while**

---

# J   Additional Experiment Results

All experiments were run on NVIDIA RTX 4090 GPUs.

| Algorithm | Test Accuracy | Time/Epoch (s) |
|---|---|---|
| Mini-batch SAM | $78.90 \pm 0.27\%$ | 13.56 |
| RW-SAM | $79.31 \pm 0.28\%$ | 15.21 |
| m-SAM ($m = 8$) | $80.72 \pm 0.12\%$ | 175.45 |
| m-SAM ($m = 16$) | $80.47 \pm 0.09\%$ | 92.22 |
| m-SAM ($m = 32$) | $80.02 \pm 0.06\%$ | 49.87 |
| m-SAM ($m = 64$) | $79.35 \pm 0.11\%$ | 26.44 |
| n-SAM | $78.15 \pm 0.19\%$ | — |

| Algorithm | Test Accuracy | Time/Epoch (s) |
|---|---|---|
| Mini-batch USAM | $78.94 \pm 0.45\%$ | 12.98 |
| m-USAM ($m = 8$) | $80.66 \pm 0.04\%$ | 173.77 |
| m-USAM ($m = 16$) | $80.46 \pm 0.07\%$ | 90.86 |
| m-USAM ($m = 32$) | $80.02 \pm 0.09\%$ | 47.09 |
| m-USAM ($m = 64$) | $79.16 \pm 0.04\%$ | 24.15 |
| n-USAM | $78.63 \pm 0.06\%$ | — |

Table 7: Left: performance and time cost of SAM, RW-SAM, m-SAM (with varying $m$), and n-SAM on CIFAR-100. Right: performance and time cost of USAM, m-USAM (with varying $m$), and n-USAM; with ResNet-18 on CIFAR-100.

Table 8: wall-clock time overhead of RW-SAM

|        | ResNet-18 | ResNet-50 | WideResNet-28-10 |
|--------|-----------|-----------|------------------|
| SAM    | 13.6      | 43.0      | 97.7             |
| RW-SAM | 15.2      | 50.9      | 112.4            |

We present the experimental results of applying the proposed reweighting strategy to ASAM (Kwon et al., 2021) in Table 9. The results demonstrate consistent improvements, and further investigation into the effectiveness of applying the reweighting strategy to different SAM variants remains an interesting direction for future work.

Table 9: Test accuracy (%) comparison between ASAM and RW-ASAM on CIFAR-10 and CIFAR-100.

| Method  | CIFAR-10 | | CIFAR-100 | |
|---------|-----------|-----------|-----------|-----------|
|         | ResNet-18 | ResNet-50 | ResNet-18 | ResNet-50 |
| ASAM    | $95.86 \pm 0.14$ | $96.12 \pm 0.23$ | $79.17 \pm 0.14$ | $80.27 \pm 0.33$ |
| RW-ASAM | $\mathbf{96.02} \pm 0.08$ | $\mathbf{96.43} \pm 0.17$ | $\mathbf{79.46} \pm 0.25$ | $\mathbf{80.65} \pm 0.16$ |

Table 10: Hyperparameter settings for fine-tuning DistilBERT on GLUE tasks.

| Task | CoLA | MNLI | MRPC | QNLI | QQP | RTE | SST-2 | STS-B | WNLI |
|------|------|------|------|------|-----|-----|-------|-------|------|
| Batch size | 32 | 64 | 32 | 64 | 64 | 32 | 64 | 32 | 32 |
| Learning rate | 2e-5 | 3e-5 | 2e-5 | 3e-5 | 3e-5 | 2e-5 | 3e-5 | 2e-5 | 2e-5 |
| Epochs | 8 | 3 | 8 | 3 | 3 | 8 | 3 | 8 | 8 |
| LR scheduler | Linear | | | | | | | | |
| Warmup ratio | 0.1 | | | | | | | | |
| Max sequence length | 256 | | | | | | | | |
| $\rho$ (for SAM and RW-SAM) | 0.05 | | | | | | | | |

