# OpenReview forum: "Unveiling m-Sharpness Through the Structure of Stochastic Gradient Noise"
_NeurIPS.cc/2025/Conference — NeurIPS 2025 poster_

### Official Review · Reviewer_Gf2z · 2025-07-02

**Clarity:** 3
**Significance:** 3
**Originality:** 2
**Rating:** 4
**Confidence:** 3

**Summary:**

This paper studies why smaller micro-batches in Sharpness-Aware Minimization (SAM) improve generalization and proposes an efficient optimizer that captures this effect. The main contributions are the followings:

- Develops a unified SDE-based analysis of SAM/USAM variants, linking smaller micro-batch size (m) to stronger variance-based regularization.
- Shows that the covariance of stochastic gradient noise (SGN) approximates the Hessian near minima, connecting SGN variance to sharpness.
- Demonstrates that this SGN-induced regularization explains the m-sharpness effect observed empirically.
- Proposes **Reweighted-SAM**, an efficient optimizer that assigns higher weight to samples with larger gradient norms.
- Reweighted-SAM avoids the high compute cost of small-m SAM by requiring only one extra forward pass.
- Empirically outperforms or matches SAM and m-SAM across CIFAR-10/100, ImageNet-1K, and ViT fine-tuning tasks.

**Questions:**

1. Can you derive an SDE for Reweighted-SAM or justify its regularization behavior? The current theory stops before the proposed optimizer. It’s unclear whether Reweighted-SAM introduces new drift/diffusion terms or still follows the same implicit regularization as SAM.

2. Why are recent lightweight SAM variants missing from experiments or discussion?
The paper does not compare Reweighted-SAM against other one-pass or efficient SAM variants. This omission makes it difficult to assess how competitive your method is in the broader landscape of sharpness-aware optimizers.  Given the paper’s theoretical focus, extensive experiments aren’t expected—but a brief discussion of lightweight SAM variants is important to contextualize Reweighted-SAM within existing work.

3. Can you show more evidence of implicit regularization? Currently, the only supporting number is the trace of the gradient noise covariance at convergence.

4. Can the authors clarify when SAM variants are worth the extra compute?
Reweighted-SAM adds only one forward pass, but this can still be costly at scale. Since SAM methods are generally more expensive than SGD or Adam, it would help to explain which real-world tasks justify this overhead.

**Ethical Concerns:**

["NO or VERY MINOR ethics concerns only"]

**Final Justification:**

I appreciate that some of my concerns were addressed. That said, the reweighted SAM still feels disconnected from the main theoretical contribution. I believe my current score (borderline accept) is a fair assessment given the current state of the paper.

**Limitations:**

Yes

**Quality:**

3

**Strengths And Weaknesses:**

### Strengths
- The paper provides a solid SDE-based analysis for various SAM and USAM variants, including closed-form expressions for the unnormalized case.

- It offers a clear explanation for the m-sharpness effect by linking it to the variance of stochastic gradient noise, which acts as an implicit regularizer.

- The proposed method shows how to retain the benefits of small-m SAM with only one additional forward pass.

### Weaknesses

- The proposed optimizer feels detached from the SDE analysis. The paper does not derive an SDE for Reweighted-SAM, so it is unclear whether the new method truly follows the same noise-regularization dynamics or introduces different effects.

- The paper only compares Reweighted-SAM against SGD, standard mini-batch SAM, and a few SAM variants. It omits newer, lightweight sharpness-aware optimizers such as **LookSAM** [4], **ESAM** [1], **GSAM** [5], **Momentum-SAM** [2], and **ASAM** [3]. Without these baselines, it is difficult to assess how Reweighted-SAM stacks up against the broader state of the art. I believe a comparison with some of these methods or at least a discussion on these prior works would be helpful.

- The authors do not derive an SDE for Reweighted-SAM, nor do they analyze its drift or diffusion behavior. Apart from reporting a single scalar (the trace of the SGN covariance at convergence), there are no deeper implicit-regularization diagnostics leaving it unclear whether Reweighted-SAM truly follows the same variance-regularization dynamics as SAM.

---

### References

[1] Du, J., Yan, H., Feng, J., Zhou, J. T., Zhen, L., Goh, R. S. M., & Tan, V. Y. F. (2022). Efficient Sharpness-Aware Minimization for Improved Training of Neural Networks. Proceedings of the International Conference on Learning Representations (ICLR).

[2] Becker, M., Altrock, F., & Risse, B. (2024). Momentum-SAM: Sharpness-Aware Minimization without Computational Overhead (arXiv:2401.12033). arXiv preprint.

[3] Kwon, J., Kim, J., Park, H., & Choi, I. Y. (2021). ASAM: Adaptive Sharpness-Aware Minimization for Scale-Invariant Learning of Deep Neural Networks. In Proceedings of the 38th International Conference on Machine Learning (ICML) (pp. 6060–6069).

[4] Liu, Y., Mai, S., Chen, X., Hsieh, C.-J., & You, Y. (2022). Towards Efficient and Scalable Sharpness-Aware Minimization. In Proceedings of the IEEE/CVF Conference on Computer Vision and Pattern Recognition (CVPR).

[5] Zhuang, J., Gong, B., Yuan, L., Cui, Y., Adam, H., Dvornek, N., Tatikonda, S., Duncan, J. S., & Liu, T. (2022). Surrogate Gap Minimization Improves Sharpness-Aware Training. In Proceedings of the 39th International Conference on Machine Learning (ICML) (pp. 26848–26872).

---

> ### Author Rebuttal · Authors · 2025-07-30
>
> We thank the reviewer for their detailed evaluation and constructive suggestions. Below, we address each of the points raised.
>
> **1. Can you derive an SDE for Reweighted-SAM or justify its regularization behavior?**
>
> Reweighted-SAM is essentially a heuristic algorithm inspired by our theoretical analysis. Although we can formally derive an SDE for Reweighted-SAM, the result contains highly complex coupling terms between the per-sample weights $p_i$ and the gradients, making the equation neither intuitive nor particularly informative for understanding the implicit regularization.
>
> **2. Why are recent lightweight SAM variants missing from experiments or discussion?**
>
> Our paper is primarily focused on studying the m-sharpness phenomenon in SAM and developing a theoretical framework to understand it, which is why we did not cover the full range of empirical SAM variants.  We acknowledge this oversight.
> Our data‑reweighting approach is complementary to parameter‑level variants such as ASAM and GSAM, so it can be combined with them directly. In the rebuttal, we added experiments on ASAM with reweighting (termed RW‑ASAM); the results below clearly demonstrate that our method integrates effectively with ASAM and improves performance. In the revised manuscript, we will include a discussion of how our reweighting method can be paired with other lightweight SAM variants, including LookSAM, ESAM, GSAM, Momentum-SAM, and ASAM, and present more extensive experimental results.
> ### CIFAR-10
> | Method   | ResNet-18        | ResNet-50        |
> |---------|------------------|------------------|
> | ASAM    | 95.86 ± 0.14     | 96.12 ± 0.23     |
> | RW-ASAM | **96.02 ± 0.08** | **96.43 ± 0.17** |
>
> ### CIFAR-100
> | Method   | ResNet-18        | ResNet-50        |
> |---------|------------------|------------------|
> | ASAM    | 79.17 ± 0.14     | 80.27 ± 0.33     |
> | RW-ASAM | **79.46 ± 0.25** | **80.65 ± 0.16** |
>
> **3. Can you show more evidence of implicit regularization?**
>
> In our paper, we quantify implicit regularization via the trace of the SGN covariance matrix according to our theory. Beyond the final trace values, Figures 1 and 2 in Section 3.5 plot its evolution throughout training and demonstrate a strong correlation with generalization performance. In the revised manuscript, we will include analogous learning curves for Reweighted-SAM—visually illustrating how its SGN trace evolves and induces implicit regularization over time (we are unable to upload these figures here due to NeurIPS policy this year).
>
> **4. Can the authors clarify when SAM variants are worth the extra compute?**
>
> We anticipate that in overparameterized models—where training loss can be driven to near zero—the extra cost of stronger implicit regularization is often worthwhile. In such settings, the abundance of global minima means methods like SAM and Reweighted-SAM can steer optimization toward flatter minima that generalize better (as discussed in our Section 3.5). Conversely, for tasks where convergence speed is paramount—such as LLMs—the overhead of SAM and Reweighted-SAM may outweigh their benefits.

---

> > ### Comment · Reviewer_Gf2z · 2025-08-04
> >
> > Thank you for the detailed rebuttal. I appreciate that some of my concerns were addressed. That said, the reweighted SAM still feels disconnected from the main theoretical contribution. I believe my current score (borderline accept) is a fair assessment given the current state of the paper.

---

### Official Review · Reviewer_TwRD · 2025-07-02

**Clarity:** 2
**Significance:** 3
**Originality:** 2
**Rating:** 4
**Confidence:** 2

**Summary:**

The authors explore the phenomenon of m-sharpness in SAM, where reducing the micro-batch size during perturbation leads to better generalization. Using a detailed SDE framework, they analyze various SAM variants and uncover that SGN inherently contributes to implicit sharpness regularization. Based on this insight, they propose Reweighted-SAM that adaptively samples data based on SGN magnitude to achieve strong generalization while maintaining efficiency. Extensive experiments on CIFAR and ImageNet show that Reweighted-SAM outperforms standard SAM without the high overhead of m-SAM.

**Questions:**

- I think Reweighted-SAM addresses parallelization using Monte Carlo sampling. Can you explain further in detail?
- Line 228. The orthogonality assumption between SGN and full gradient seems strong. What happens if this is not held? And does it have any effects on the parallelization characteristic of Reweighted-SAM?
- Too many proofs and maths, will you disclose your code during the review process so we can have a further check on the approach?

**Ethical Concerns:**

["NO or VERY MINOR ethics concerns only"]

**Final Justification:**

I have no further questions after the authors' responses. I will not change my rating.

**Limitations:**

yes

**Quality:**

3

**Strengths And Weaknesses:**

**Strengths**
- Solid theoretical framework using advanced SDE analysis to explain m-sharpness.
- Strong empirical results across datasets and models, including robustness to label noise.

**Weaknesses**
- Theoretical analysis is dense and may lack accessibility for a broader reader group.
- No ablation comparing Reweighted-SAM directly with other SGN-regularization techniques or alternative reweighting schemes.

---

> ### Author Rebuttal · Authors · 2025-07-30
>
> We thank the reviewer for their thoughtful evaluation. Below we address each of the concerns raised.
>
> **1. Theoretical analysis is dense and may lack accessibility for a broader reader group.**
>
> In the revised version, we will reduce the density of formulas in the theory section, add more intuitive explanations to improve readability, and provide additional background on SDEs to make the content easier for readers to follow.
>
> **2. No ablation comparing Reweighted-SAM directly with other SGN-regularization techniques or alternative reweighting schemes.**
>
> To our knowledge, ours is the first algorithm to incorporate a reweighting mechanism into SAM. Existing SGN-based regularization methods typically struggle to match SAM’s performance and cannot be directly integrated into the SAM framework.
>
> **3. I think Reweighted-SAM addresses parallelization using Monte Carlo sampling. Can you explain further in detail?**
>
> We assign each sample an importance weight related to its SGN norm, and use Monte Carlo sampling to estimate these per-sample gradient norms with only a single forward pass over the minibatch. This MC-based weight estimation is the key enabler that makes our approach fully parallelizable.
>
> **4. Line 228. The orthogonality assumption between SGN and full gradient seems strong. What happens if this is not held? And does it have any effects on the parallelization characteristic of Reweighted-SAM?**
>
> The orthogonality between the full gradient and noise is merely an intuitive assumption. In fact, we only require that the stochastic‐gradient norm closely approximates the noise norm, which empirically holds in mid-to-late training. For instance, on CIFAR‑100 with ResNet‑18, we observed that, for most samples, the ratio of noise norm to gradient norm exceeds 0.98. While we have not conducted a dedicated robustness study on this orthogonality assumption, our noisy‑label experiments (Table 3) implicitly stress it: injected label noise strongly violates any near‑orthogonality, yet Reweighted‑SAM still outperforms all baselines. In particular, at an 80% noise ratio, Reweighted‑SAM achieves a 16% absolute accuracy gain. These results indicate that our method does not critically depend on the SGN–gradient orthogonality assumption. The orthogonality does not compromise the algorithm’s parallelizability.
>
> **5. Too many proofs and maths, will you disclose your code during the review process so we can have a further check on the approach?**
>
> Due to this year’s NeurIPS policy, we are unable to upload code during the rebuttal period; however, we will be happy to release our implementation if the paper is accepted.

---

> > ### Comment · Reviewer_TwRD · 2025-08-05
> >
> > Thank you for your response. I still have a few questions.
> >
> > Three USAM variants and three vanilla SAM variants were mentioned in your contributions. Why were the experimental results for these methods not presented? What makes them impractical, if that’s the case?
> >
> > How sensitive is RW-SAM to the sample size parameters since you are using MC methods?
> >
> > The paper states, "Foret et al. (2021) have shown that SAM exhibits robustness to label noise," but the results in your Table 3 show that SAM performs significantly poorly under the highest noise conditions.
> >
> > Can RW-SAM be directly applied to training in regression problems? If so, does the noise robustness property also hold in that context?

---

> > > ### Author Response · Authors · 2025-08-05
> > >
> > > Thank you for your response and follow-up questions. Below we address each point in turn.
> > >
> > > **1. Three USAM variants and three vanilla SAM variants were mentioned in your contributions. Why were the experimental results for these methods not presented? What makes them impractical, if that’s the case?**
> > >
> > > Due to space constraints, we have deferred the empirical performance and computational overhead of all six variants—evaluated on ResNet-18 trained on CIFAR-100—to Table 6 in Appendix J, as noted in the main paper. We did not conduct extensive experiments on the n-SAM/USAM and m-SAM/USAM variants because their computational overhead is prohibitively large due to their non-parallelizability; this in fact motivated our proposal of RW-SAM.
> > >
> > > **2. How sensitive is RW-SAM to the sample size parameters since you are using MC methods?**
> > >
> > > Our empirical observations show that using an MC sample size of 1 leads to roughly a 40% relative error in gradient-norm estimation, whereas increasing it to 10 reduces the error to about 20%. However, because this incurs ten times the training overhead for only marginal performance gains (~0.1%), we use an MC sample size of 1 in all experiments to maximize efficiency.
> > >
> > > **3. The paper states, "Foret et al. (2021) have shown that SAM exhibits robustness to label noise," but the results in your Table 3 show that SAM performs significantly poorly under the highest noise conditions.**
> > >
> > > Your observation is correct. This indicates that SAM’s choice of $\rho$ may not be sufficiently robust under high noise ratios. In the paper by Foret et al. (2021), they also note that at an 80% noise ratio, additional tuning of $\rho$ is needed, whereas we used the default $\rho$ value across all noise ratios. Even so, RW-SAM still substantially outperforms both SAM and SGD, despite SAM’s poor performance in the highest noise condition.
> > >
> > > **4. Can RW-SAM be directly applied to training in regression problems? If so, does the noise robustness property also hold in that context?**
> > >
> > > RW-SAM can in principle be applied to regression, as neither our theoretical analysis nor its formulation relies on a specific loss function. However, we have not explored this setting in the present work. We have followed the standard evaluation protocol for SAM variants, and to our knowledge few studies apply SAM to regression tasks. We leave testing RW-SAM’s performance and its robustness to label noise in regression settings to future work.

---

### Official Review · Reviewer_19PB · 2025-07-03

**Clarity:** 3
**Significance:** 3
**Originality:** 3
**Rating:** 5
**Confidence:** 3

**Summary:**

The paper investigates the *m-sharpness* phenomenon in Sharpness-Aware Minimization (SAM), where generalization improves as the micro-batch size used for perturbation decreases. Extending the stochastic differential equation (SDE) framework to jointly model learning rate and perturbation magnitude, the authors provide theoretical insight into how stochastic gradient noise (SGN) induces implicit regularization in SAM and its variants. Motivated by this analysis, they propose Reweighted-SAM, an algorithm that adaptively reweights samples based on estimated SGN magnitude. This method aims to reproduce the benefits of m-SAM while remaining parallelizable. Experiments on standard vision benchmarks support the method’s efficacy.

**Questions:**

* How sensitive is Reweighted-SAM to violations of the SGN orthogonality assumption? Have the authors empirically studied its robustness under such conditions?
* Could Reweighted-SAM be adapted to other sharpness-aware variants such as ASAM or FSAM? What adjustments would be necessary?
* How critical is the SGN reweighting parameter λ in practice? Does it require tuning across datasets, or is normalization sufficient to ensure stable performance?
* Have the authors explored ways to reduce the overhead of SGN magnitude estimation, such as computing it less frequently or using alternative proxies?
* To what extent does Reweighted-SAM generalize beyond supervised image classification? Have the authors considered applications in NLP or reinforcement learning?

**Ethical Concerns:**

["NO or VERY MINOR ethics concerns only"]

**Final Justification:**

I have raised my score (from 4 to 5) based on the authors' strong rebuttal, which resolved my initial concerns. They provided new experiments demonstrating broader applicability to other optimizers and domains and clarified that their theoretical framework does not strictly require the strong assumptions I had questioned. The paper's core contributions—providing novel theoretical insight into m-sharpness and proposing a new algorithm—are significant. The computational overhead is a minor limitation but represents a reasonable trade-off for the performance gains and improved efficiency over m-SAM. The paper's strengths now clearly outweigh its weaknesses.

**Limitations:**

Yes.

**Paper Formatting Concerns:**

None.

**Quality:**

3

**Strengths And Weaknesses:**

*Strengths*

* The paper offers a rigorous and well-motivated theoretical framework that explains the role of SGN in implicit sharpness regularization.
* Reweighted-SAM is a novel algorithm derived directly from the theoretical findings and is more computationally efficient than m-SAM.
* Empirical results are thorough and demonstrate consistent gains across various datasets, architectures, and settings.
* The analysis provides a compelling explanation for the generalization gap between full-batch and micro-batch SAM variants.

*Weaknesses*

* The theoretical analysis depends on strong assumptions, including orthogonality between SGN and the full gradient, and properties such as log-concavity and Lipschitz continuity, which may not hold in practice.
* Some sections, especially those involving the extended SDE derivations, are dense and may be difficult for readers without a strong mathematical background.
* Although Reweighted-SAM improves over m-SAM in efficiency, it still introduces nontrivial computational overhead, and the scalability to large models remains unclear.
* Baseline comparisons omit several recent or stronger alternatives in sharpness-aware optimization, limiting the empirical context of the results.

---

> ### Author Rebuttal · Authors · 2025-07-30
>
> We thank the reviewer for the insightful comments and address the concerns point by point below.
>
> **1. The theoretical analysis depends on strong assumptions, including orthogonality between SGN and the full gradient, and properties such as log-concavity and Lipschitz continuity, which may not hold in practice.**
>
> We would like to clarify that these assumptions are used in different contexts rather than all at once. Specifically: Lipschitz continuity is assumed only to guarantee the existence and uniqueness of a strong solution to the SDE; if weak solutions are acceptable, this requirement can be relaxed. Log‑concavity of the gradient distribution appears only in the extra statement of Proposition 3.9 to ensure monotonicity with respect to batch size; even without log‑concavity, the main proposition still holds. The orthogonality between the full gradient and noise is merely an intuitive assumption and is not required by our core theory. In practice, we only need the norm of the stochastic gradient to approximate the norm of the gradient noise—which is a good approximation for USAM or in late‐training when the full‐batch gradient norm is small—so strict orthogonality is unnecessary.
>
> **2. Some sections, especially those involving the extended SDE derivations, are dense and may be difficult for readers without a strong mathematical background.**
>
> In the revised version, we will reduce the density of formulas in the theory section, add more intuitive explanations to improve readability, and provide additional background on SDEs to make the content easier for readers to follow.
>
> **3. Could Reweighted-SAM be adapted to other sharpness-aware variants such as ASAM or FSAM? What adjustments would be necessary?**
>
> Our data‑reweighting approach is complementary to parameter‑level variants such as ASAM and FSAM, so it can be combined with them directly. In the rebuttal, we added experiments on ASAM with reweighting (termed RW‑ASAM); the results below clearly demonstrate that our method integrates effectively with ASAM and improves performance. In the revised manuscript, we will include a discussion of how our reweighting method can be paired with other SAM variants and present more extensive experimental results.
>
> ### CIFAR-10
> | Method   | ResNet-18        | ResNet-50        |
> |---------|------------------|------------------|
> | ASAM    | 95.86 ± 0.14     | 96.12 ± 0.23     |
> | RW-ASAM | **96.02 ± 0.08** | **96.43 ± 0.17** |
>
> ### CIFAR-100
> | Method   | ResNet-18        | ResNet-50        |
> |---------|------------------|------------------|
> | ASAM    | 79.17 ± 0.14     | 80.27 ± 0.33     |
> | RW-ASAM | **79.46 ± 0.25** | **80.65 ± 0.16** |
>
> **4. How sensitive is Reweighted-SAM to violations of the SGN orthogonality assumption? Have the authors empirically studied its robustness under such conditions?**
>
> As noted in our response to Point 1, strict orthogonality between the full gradient and noise is not required—what matters is that the norm of the stochastic gradient closely approximates the norm of the noise, which empirically holds in mid‑to‑late training. For instance, on CIFAR‑100 with ResNet‑18, we observed that, for most samples, the ratio of noise norm to gradient norm exceeds 0.98. While we have not conducted a dedicated robustness study on this orthogonality assumption, our noisy‑label experiments (Table 3) implicitly stress it: injected label noise strongly violates any near‑orthogonality, yet Reweighted‑SAM still outperforms all baselines. In particular, at an 80% noise ratio, Reweighted‑SAM achieves a 16% absolute accuracy gain. These results indicate that our method does not critically depend on the SGN–gradient orthogonality assumption.
>
> **5. How critical is the SGN reweighting parameter λ in practice? Does it require tuning across datasets, or is normalization sufficient to ensure stable performance?**
>
> As demonstrated in Section 5.4 and Table 4, our algorithm is largely insensitive to the choice of λ: the default setting is generally robust, thanks to the normalization step in our method.
>
> **6. Have the authors explored ways to reduce the overhead of SGN magnitude estimation, such as computing it less frequently or using alternative proxies?**
>
> We did not explore this in our current implementation because we estimate SGN magnitude on a per‑sample basis, and reducing the estimation frequency risks producing overly coarse reweighting—especially on large datasets. Exploring techniques to mitigate the extra overhead of our reweighting scheme is a direction for future work.
>
> **7. To what extent does Reweighted-SAM generalize beyond supervised image classification? Have the authors considered applications in NLP or reinforcement learning?**
>
> During the rebuttal period, we added experiments on three GLUE tasks using DistilBERT with AdamW as the base optimizer trained for 10 epochs, and report the median results over five independent runs below. We find that, although SAM sometimes underperforms (e.g., on RTE), RW‑SAM consistently improves upon SAM. In the revised manuscript, we will include results for the full GLUE benchmark.
>
> | Method   | SST‑2 | RTE    | STS‑B (Pearson / Spearman) |
> |----------|-------|--------|----------------------------|
> | AdamW    | 90.7  | **63.9** | 86.9 / 86.8                |
> | SAM      | 91.3  | 61.4   | 87.0 / 86.9                |
> | RW‑SAM   | **91.7** | 62.5   | **87.2 / 87.0**              |
>
> In reinforcement learning, there have been very recent efforts to adapt SAM (Lee & Yoon, 2025), but it is not yet popular in RL. Since RL setups differ fundamentally from supervised learning, developing a rigorous theoretical framework for SAM in RL will require additional technical advances and represents a promising direction for future work.
>
> Lee, H. K., & Yoon, S. W. (2025, April). Flat reward in policy parameter space implies robust reinforcement learning. In The Thirteenth International Conference on Learning Representations.

---

> > ### Comment · Reviewer_19PB · 2025-08-06
> >
> > To the authors,
> > Thank you for the excellent and thorough rebuttal. The additional experiments for ASAM and GLUE were highly convincing, and I appreciate the detailed clarifications on the theoretical assumptions. Your responses have successfully addressed my major concerns. This is a solid contribution to the field.

---

### Official Review · Reviewer_vX9u · 2025-07-03

**Clarity:** 3
**Significance:** 3
**Originality:** 3
**Rating:** 4
**Confidence:** 4

**Summary:**

This paper provides a theoretical and empirical investigation into m-sharpness, the phenomenon that Sharpness-Aware Minimization (SAM)  improves generalization more as the micro-batch size $m$ used for perturbation decreases (i.e. using more smaller disjoint batches for the SAM inner maximization). The authors extend the stochastic differential equation (SDE) framework for analyzing stochastic gradient descent (SGD) to two parameters (the learning rate $\eta$ and SAM perturbation radius $\rho$) and derive continuous-time approximations for several SAM variants. Building on these approaximations, the paper’s contributions are: (1) a rigorous SDE-based analysis of SAM, revealing a variance-driven implicit sharpness regularization mechanism underlying m-sharpness; (2) a unifying comparison of SAM, m-SAM, and related variants (normalized vs. unnormalized) in terms of their dynamics; and (3) a new sharpness-weighted data sampling technique (Reweighted SAM) inspired by the theory, with empirically demonstrated efficacy.

**Questions:**

Please see the weakness section.

could u authors comment on exact technical challenges involved in extension to two-parameter weak approximation. Prima-facie it looks a simple enough natural extension following up on previous works such as compognoni 2023 but i will reserve my judgement.

**Ethical Concerns:**

["NO or VERY MINOR ethics concerns only"]

**Final Justification:**

I will stick to my rating. I dont have major concerns about the paper but because of its relatively incremental nature i dont see many discussions happening around this paper at the conference which is why I am hesistant to give a higher rating.

**Limitations:**

the authors have mentioned computation overhead, but this paragraph could have been expanded with additional comments about other variants of sam and RWSAM's extension to those.

**Quality:**

3

**Strengths And Weaknesses:**

Strengths:

The paper is technically rigorous and the mathematical derivations in the paper appear sound and build upon established theory for interpreting SGD as a stochastic differential equations. The authors formalize a two-parameter weak convergence expansion (in Appendix A) that allows $\eta \to 0$ and $\rho \to 0$ at independent rates. This generalization of prior one-parameter analyses (e.g. Compagnoni et al. 2023and Luo et al. 2025) is non-trivial and is executed with rigor. In particular, Theorem 3.3 (for mini-batch USAM) and Theorem 3.5 (for m-USAM) are derived by applying Dynkin’s formula and carefully controlling error terms $O(\eta^{\alpha}\rho^{\beta})$. The resulting SDEs are also  intuitively reasoanble. The key finding – that stochastic gradient noise introduces a $\nabla \operatorname{tr}(V)$ regularization term – is a deep insight that advances our understanding of why SAM (especially with small $m$) finds flatter minima.

Additional algorithmic contribution — Rather than just providing post-hoc analysis, the authors use the theory to inspire a new method. RW-SAM is conceptually simple (importance-sampling the batch for perturbation based on gradient norm) yet effective. It addresses the key limitation of m-SAM (lack of parallelization) by staying with a single mini-batch but simulating the effect of smaller $m$. This idea is novel, and the experiments show it works: RW-SAM consistently outperforms standard SAM by a small but non-negligible margin (e.g., +0.4–0.5% on CIFAR-100, see Table 1) with only ~10-20% training overhead, and is much faster than true m-SAM (which requires serial micro-batch updates).

The empirical results are interesting and demonstrate that RW SAM consisntely outperforms vanilla SAM, though the improvements seem minor compared to more recent variations of SAM.

The paper is clearly written and well-organized. The theoretical results with their assumptions, development of new results and their context with known results from previous papers are all well presented.

Weaknesses:

The related works are largely well-cited except a couple of important misses such as Keskar et al. (2017), who originally identified the connection between batch size, sharpness, and generalization. And the Vasso paper by Li and Giannakis which is also very relevant. mSam explorations by Behdin et al seems also relevant (msam: Micro-batch-averaged sharpness-aware minimization)

Some of the dense sections could be written better and made more accessible but this is a highly technical paper and warrants some background e.g. in stochastic calculus to fully understand and appreciate. i have not gone through all the proofs etc.


Some of the language could be toned down as it reads as over claiming e.g. the gradient variance is certainly important  as found by the analysis but would not fully explain the generalization. . It might be worth explicitly acknowledging that m-sharpness holds empirically in many cases but not universally (they cite a result in Wen et al. 2022 about transformers where it fails).

The empirical section could have been expanded to include more varied domains, model architectures and settings.

Minor :

Theorem 3.4 — incorrect equation is references (7) instead of (10)

Line 289 : “tables” -> table.

There are some grammar issues .e.g “Motivated by this, we  evaluate RW-SAM maintains”

---

> ### Author Rebuttal · Authors · 2025-07-30
>
> We thank the reviewer for the careful reading and positive feedback. Below we address the specific points raised:
>
> **1. Could u authors comment on exact technical challenges involved in extension to two-parameter weak approximation.**
>
> A key technical challenge lies in our need to simultaneously track both $\eta$ and $\rho$ in Dynkin expansion (Lemma A.3), and to develop a matched two‑parameter moment argument for both the continuous SDE and the discrete update (Lemmas A.4–A.5). In doing so, we must carefully control all mixed remainder terms uniformly for any relative scaling $\eta,\rho\to0$, which is more complex than the single‑parameter case. Compared to prior SDE analyses for SAM (e.g., Compagnoni et al., 2023), which directly invoke the weak‑approximation result for $\eta$ from Li et al. (2019), we avoid the combinatorial explosion of cross‑terms arising from an Itô–Taylor expansion (see Lemma 28 in Li et al. 2019) by using our generator‑based approach—resulting in a more concise derivation and a precise quantification of each order’s contribution.
>
> **2. Missing related work.**
>
> We have already cited Behdin et al. (2023). In the revised manuscript, we will also include discussion and citations for Keskar et al. (2017) and Li & Giannakis (2023).
>
> **3. Tone and language adjustments.**
>
> We will soften overly strong claims about generalization, explicitly acknowledge the empirical limits of m-sharpness, reduce the density of formulas in the theory section, and add more intuitive explanations to improve readability, as suggested by the reviewer.
>
> **4. Minor corrections.**
>
> We apologize for these errors and will correct them in the revised manuscript.
>
> **5. The empirical section could have been expanded to include more varied domains, model architectures and settings.**
>
> During the rebuttal period, we added experiments on three GLUE tasks using DistilBERT with AdamW as the base optimizer trained for 10 epochs, and report the median results over five independent runs below. We find that, although SAM sometimes underperforms (e.g., on RTE), RW‑SAM consistently improves upon SAM. In the revised manuscript, we will include results for the full GLUE benchmark.
>
> | Method   | SST‑2 | RTE    | STS‑B (Pearson / Spearman) |
> |----------|-------|--------|----------------------------|
> | AdamW    | 90.7  | **63.9** | 86.9 / 86.8                |
> | SAM      | 91.3  | 61.4   | 87.0 / 86.9                |
> | RW‑SAM   | **91.7** | 62.5   | **87.2 / 87.0**              |
>
> **6. RWSAM's extension to other variants of sam.**
>
> We have already obtained preliminary experimental results by applying our proposed reweighting scheme to other SAM variants, and we will include a discussion of these combined approaches in the revised manuscript.

---

> > ### Comment · Reviewer_vX9u · 2025-08-06
> >
> > Thank you, I'll stick to my score.

---

### Decision · Program_Chairs · 2025-09-17

**Decision:**

Accept (poster)

**Comment:**

The authors investigate sharpness-aware minimization (SAM) via a phenomenon known as m-sharpness where the performance of SAM improves monotonically as the micro-batch size for computing perturbations decreases. The authors use a SDE framework to analyze this phenomenon. They find that the stochastic noise introduced during SAM perturbations inherently induces a variance-based sharpness regularization effect. The paper then introduces a reweighted SAM.

All the reviewers were positive about the paper. They thought the results were theoretically rigorous and interesting. Several reviewers increased their scores after the rebuttal from the authors. Some additional work cited needs to be included as well as a few additional experiments that appeared to be performed during the rebuttal.